# PICL: Incorporating Coarse-Grained Data and Physics Information for Superior Physical Systems Modeling

## Abstract

Physics-informed machine learning has emerged as a promising approach for modeling physical systems. However, two significant challenges limit its real-world applicability. First, most realistic scenarios allow only coarse-grained measurements due to sensor limitations, making the use of physics loss based on finite dimensional approximations infeasible. Second, the high cost of data acquisition impedes the model's predictive ability. To address these challenges, we introduce a novel framework called Physics-Informed Coarse-grained data Learning (PICL) that incorporates physics information via the learnable fine-grained state representation from coarse-grained data. This framework effectively integrates data-driven methods with physics-informed objectives, thereby significantly improving the predictive ability of the model to predict the subsequent coarse-grained observations from current coarse-grained observation. The PICL framework comprises two modules: the encoding module, responsible for generating the learnable fine-grained state, and the transition module, used for predicting the subsequent state. To train these modules, we employ a base-training period followed by a two-stage fine-tuning period. The key idea behind this training strategy is that we can leverage physics loss to enhance the reconstruction ability of the encoding module and the generalization ability of the transition module, using both labeled and unlabeled data. In the base-training period, we train both modules collaboratively using data loss and physics loss. In the two-stage fine-tuning period, we first tune the transition module with physics loss using unlabeled data and then tune the encoding module with data loss using labeled data to propagate the information from the transition module to the encoding module. We demonstrate that PICL exhibits superior predictive ability across modeling various PDE-governed physical systems. Code is available on GitHub: https://github.com/PI-CL/PICL.

## 1 Introduction

Physical systems modeling is often used to approximate complex natural phenomena (Wang et al., 2023). To leverage it as the surrogate for forward prediction and inverse design, the models of physical systems must accurately predict the future of the system. Using neural networks to approximate the physical systems by using data-driven methods has become a promising direction (Zhang et al., 2023). Recently, to reduce the use of costly data and improve the predictive ability of physical system models, several works have introduced finite-dimensional approximated physics loss in training models and achieved great performance (Gao et al., 2021a; Ren et al., 2022; Huang et al., 2023). Many methods rely on fine-grained data to calculate physics loss, but in some scenarios, we only have insufficient data on the coarse-grained mesh (Watson et al., 2020). This introduces significant errors in computing partial derivatives, which hurts the accuracy of the physics loss. For example, ocean circulation, governed by the shallow water equation, affects the pollutant transfer (Hu et al., 2020). However, we can only measure the values of flow velocity and pollutants in sparse fixed monitors (Bas et al., 2019; Su et al., 2023). As a result, enhancing neural networks trained by coarse-grained data using known physics equations remains a scientific challenge.

In this work, we introduce a novel Physics-Informed Coarse-grained data Learning framework, known as PICL, that integrates physics information into the training of the model when we only

have the coarse-grained data. PICL comprises an encoding module and a transition module. The fundamental concept is to reconstruct the learnable fine-grained state from the coarse-grained input by using the encoding module and then predict the subsequent state with the transition module. However, the encoding module makes it hard to generate the fine-grained state without available fine-grained data through data-driven supervised methods, and if the training data is insufficient, the model cannot accurately predict the system's future prediction. To address these challenges, we propose jointly training the encoding and transition modules by utilizing physics losses and data loss, along with both limited labeled and richer unlabeled coarse-grained data, to predict the subsequent coarse-grained observation.

To train both modules, we propose a training framework comprising two periods: a base-training period and a two-stage fine-tuning period. In the base-training period, the encoding module is trained using a physics loss calculated based on the PDEs formulation without requiring fine-grained data. Drawing inspiration from the PDEs formulation and finite difference method (FDM), where there exists an equivalence relation between temporal and spatial differences, allowing for interconversion between them, we use the most recent consecutive observations to generate a more reliably learnable fine-grained state. The transition module is trained collaboratively using data loss and physics loss to overcome the limited predictive ability caused by insufficient data. During the two-stage fine-tuning period, the model can be further improved by using more available unlabeled data in a semi-supervised learning way. In the first stage, the transition module is fine-tuned using unlabeled data and physics loss independently. In the second stage, we employ data loss based on the original labeled data to fine-tune the encoding module independently. By doing so, we propagate the information of the PDEs and unlabeled data from the transition module to the encoding module.

We demonstrate the effectiveness of PICL in three different PDEs, e.g., wave equation, linear shallow water equation, and nonlinear shallow water equation with uneven bottom. We find that, with PICL, the learned model predicts the future coarse-grained observation more accurately.

Our contributions can be summarized in two parts: **(1)** We propose a general physics-informed deep learning framework called PICL integrating physics information into the training of model if we only have coarse-grained data. **(2)** We demonstrate that PICL leads to a significant improvement in predictive ability to predict the subsequent coarse-grained observations with input coarse-grained observation at time $t$, than data-driven manner across various PDE-governed physical systems.

## 2 RELATED WORK

**Physics-Informed Neural Operator**: There are two categories to train deep models of PDEs dynamics. The first one is the data-driven method using the data set collected from solvers or experiments, like the neural operators (Lu et al., 2019; Li et al., 2020a;b; Boussif et al., 2022; Yin et al., 2023; Iakovlev et al., 2023) learn an operator including a family of parametric PDEs, instead of only a function. Another one is Physics-Informed Neural Networks (PINNs) (Raissi et al., 2019; Yang et al., 2021; Cai et al., 2021; Karniadakis et al., 2021) for training PDEs constrained loss to solve an equation. Both approaches have disadvantages. On the one hand, neural operators require data, and when data is sparse, insufficient, or not available, they cannot learn the solution operator successfully. However, data generation might require the enormous cost of the expensive solver and experiment. On the other hand, PINNs do not mandate the input of data, which tends to exhibit limitations, particularly in the context of multi-scale dynamic systems, attributable to the complexities of optimization (Rao et al., 2023). In addition, PINNs have the problem of slow calculation compared with conventional CFD like Finite Element Method (FEM) (Reddy, 2019), Lattice Boltzmann Method (IBM) (Chen & Doolen, 1998) and Boundary Data Immersion Method (BDIM) (Weymouth & Yue, 2011). In order to overcome the above challenges, physics-informed operator learning has been proposed in DeepONet (Wang et al., 2021; Goswami et al., 2022) and in FNO (Li et al., 2021) that reduce the need for data by injecting physics information and learn an operator to generalize multi-scale dynamics. However, the practical implementation of the above methods often still requires high-quality and complete data, and they cannot be applied to learn coarse-grained measured data directly as opposed to low-resolution data. We develop PICL to overcome these problems further so that the physics loss can be integrated to learn coarse-grained observations.

**Super-Resolution**: Super-resolution (SR) represents a fundamental task in low-level vision, aiming to reconstruct a high-resolution (HR) image from its low-resolution (LR) counterpart. HR tasks

are focused on two domains: computer vision (CV) and physical systems. In CV, the pioneering study in Dong et al. (2014) was among the first to utilize deep learning for SR. Subsequent to this, numerous deep learning-based models (Lim et al., 2017; Soh et al., 2019; Nazeri et al., 2019; Zhao et al., 2020), and generative models (Ledig et al., 2017; Liu et al., 2021; Gao et al., 2023) emerged to enhance SR performance. SR for physical systems has attracted more and more attention (Ren et al., 2022), which aims to inject physics information into the SR model (Wang et al., 2020; Esmaeilzadeh et al., 2020; Fathi et al., 2020; Ren et al., 2022; Jangid et al., 2022; Shu et al., 2023) or develop the physics-informed SR models without data (Gao et al., 2021b; Kelshaw et al., 2022; Zayats et al., 2022). Both methods face challenges in terms of data requirements and predictive ability despite their individual merits. On the one hand, SR in CV and most of the work in physical systems require HR data used in supervised learning. On the other hand, other works in physical systems only rely on physics information, often failing to provide the expected HR reconstructions, especially when LR data makes up only a small fraction. Our proposed PICL first applies physics-informed training to learn the HR state via coarse-grained input without HR data and embeds it to a prediction task further instead of using SR as an end-to-end task like the above works.

## 3 PRELIMINARIES

Consider the dynamical system following the work (Li et al., 2021):

$$
\begin{aligned}
\frac{du}{dt} &= \mathcal{P}(u), && \text{in } \Omega \times (0, \infty), \\
u &= g, && \text{in } \partial\Omega \times (0, \infty), \\
u &= a, && \text{in } \bar{\Omega} \times \{0\},
\end{aligned}
\tag{1}
$$

where the unknown solutions $u(t) \in \mathcal{U}$ for $t > 0$ and the initial conditions (ICs) $a = u(0) \in \mathcal{A} \subseteq \mathcal{V}$. The operator $\mathcal{P}$, which possibly be a linear or non-linear partial differential operator with $\mathcal{U}$ and $\mathcal{V}$ in Banach spaces. $g$ denotes the known boundary conditions (BCs). It is hypothesized that $u$ is both existent and bounded at all times, applicable to every $u_0 \in \mathcal{U}$.

As the inherent complexity of PDEs, a large portion of them rely on numerical methods to solve Eqn. 1 (Shi et al., 2022). Numerical solvers operate by transforming PDEs into algebraic equations via discretization and then undertaking their numerical solution. In this process, $a$ is discretized at $n_a$ points, producing $\tilde{a}$, and $u$ is represented by $\tilde{u}$ discretized at $n_u$ points. Consequently, $\tilde{\mathcal{P}}$ is an algebraic equations to approximate $\mathcal{P}$. We first solve $\tilde{\mathcal{P}}$ to generate $\tilde{u}$ on fine-grained mesh and appropriate $\tilde{a}$. We down-sample the fine-grained resolution to the coarse-grained resolution, denoted as $\tilde{o}$. The detail of down-sampling is introduced in Appendix B.5. Then, we get the coarse-grained data set $\mathcal{D} = \{\tilde{o}_i\}_{i=0}^{N}$. Except for the insufficient labeled data, the unlabeled data which is the current coarse-grained observation without the corresponding subsequent observation, can also be applied effectively in our proposed PICL. We denote the unlabeled data set as $\mathcal{B} = \{\tilde{o}_i^{\eta}\}_{i=0}^{N'}$.

## 4 PHYSICS-INFORMED COARSE-GRAINED DATA LEARNING

Here we describe the overview of the proposed methodology (Sec. 4.1) and the strategy for implementation (Sec. 4.2) including the framework (Sec. 4.2.1) and the learning strategy (Sec. 4.2.2).

### 4.1 OVERVIEW OF METHODOLOGY

As we mentioned in the introduction, we will face a challenge when we try to integrate a data-driven model based on coarse-grained data with physics information. The coarse-grained data cannot be applied to calculate physics loss directly as they only have incomplete information, while we do not have the fine-grained data to solve the super-resolution problem. Thus, the goal of this work is to develop a framework that integrates physics information into the training of models based on coarse-grained data; when we only have coarse-grained data, we still apply physics information to help the training of models, whether in simulation or real-world application.

The fundamental idea is to reconstruct the learnable fine-grained state from the coarse-grained input by using the encoding module and then predict the subsequent state by using the transition module,

shown in Fig. 1. The aims of the two modules are to use physics loss to reconstruct the learnable fine-grained state without fine-grained data and to inject physics information using the learnable fine-grained state to improve predictive ability affected by insufficient data, respectively. To train both modules, we design two training periods: a base-training period and a two-stage fine-tuning period. In the base-training period, the encoding module is trained with a physics loss in the absence of fine-grained data. Concurrently, the transition module is trained using a combination of data loss and physics loss, addressing the challenges of limited predictive capabilities due to data scarcity. The two-stage fine-tuning period utilizes additional unlabeled data in a semi-supervised manner for further model enhancement. The first stage involves fine-tuning the transition module independently with physics loss calculated based on the unlabeled data. In the second stage, the encoding module is independently fine-tuned using data loss, with the original labeled coarse-grained data. By doing so, we propagate the information of physics and unlabeled data from the transition module to the encoding module. The details of the framework and learning strategy are described in next section.

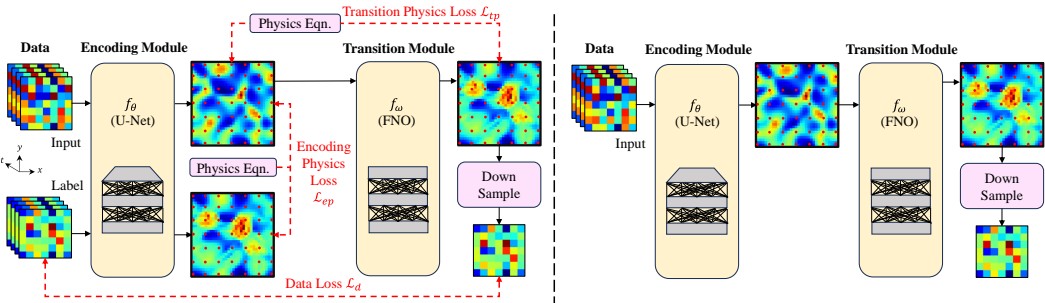

Figure 1: PICL. Base-training period (left): the encoding module is trained with a physics loss without available fine-grained data, and the transition module is trained with a combination of data loss and physics loss. Inference Period (right): given a coarse-grained observation to predict the subsequent coarse-grained observation.

## 4.2 IMPLEMENTATION STRATEGY

### 4.2.1 FRAMEWORK OF PICL

In the encoding module, PICL encodes the input $\tilde{o}_t$ to $\hat{u}_t$. Inspired by the high-order FDM that uses data from several previous steps to estimate the derivative at current time step, leading to a more accurate solution of the following state, before the data input to the encoding module, we use the abundant temporal feature of $\{\tilde{o}_{t-i}\}_{i=0}^n$ to approximate the more reliable $\hat{u}_t$ with the neural network:

$$\hat{u}_t = f_\theta(\tilde{o}_{t-n}, \tilde{o}_{t-n+1}, ..., \tilde{o}_t), \tag{2}$$

where $\theta$ is the trainable parameters and $\hat{u}_t$ is the learnable fine-grained state. We eliminate the uncertainty caused by partial observation via containing previous steps in input, which can also be handled by Bayesian neural networks (Louizos & Welling, 2017) and VAE (Kingma & Welling, 2013), but it is not the focus of this paper. Then, we use a transition module for the prediction task. The module inputs current fine-grained state $\hat{u}_t$, and predicts the subsequent fine-grained state $\hat{u}_{t+1}$. To achieve this, we use a neural network to learn the mapping of a given $\hat{u}_t$ to the subsequent $\hat{u}_{t+1}$:

$$\hat{u}_{t+1} = f_\omega(\hat{u}_t), \tag{3}$$

where $\omega$ denotes the trainable parameters in the transition module. Finally, the predicted $\hat{o}_{t+1}$ can be calculated by down-sampling with the known coordinates set $\Phi$.

In practical implementation, the encoding process is similar to the image processing task, which aims to extract features from coarse-grained input. We employ the U-Net (Ronneberger et al., 2015) as the encoding module, a proven model well-suited for tasks akin to SR. $\hat{u}_t(x,y)$ denotes the value at coordinates $(x,y)$ of the $n \times n$ matrix $\hat{u}_t$. We define a set $\Phi = \{(x,y) \mid x,y \in \{0, k, 2k, \ldots, mk\}\}$, including the coordinates of every $k$-th value in the $\hat{u}_t(x,y)$, where $m$ is a multiple of $k$. Applying the hard encoding, the $\hat{u}_t$ is modified as: $\hat{u}_t(x,y) = \{\hat{o}_t(x/k, y/k) \mid (x,y) \in \Phi\}$. For the transition

process, which can be conceptualized as a system's temporal evolution, we leverage the FNO (Li et al., 2020b) as the transition module to more effectively capture continuous and global spatial features, particularly when data availability is insufficient. Note that above architectures are choices of our work's specific requirements. Future work could explore the efficacy of alternative neural network for them.

### 4.2.2 Learning Strategy

Based on the proposed PICL framework, there are three problems to train them jointly with data loss and physics loss. The first is how to train the encoding module when we do not have the fine-grained data required by supervised learning. The second is how to leverage physics loss to improve predictive accuracy. The third is how to use unlabeled data to improve the predictive ability of physical systems modeling further. To solve the above problems, we propose a learning strategy with two periods, including a base-training period and a two-stage fine-tuning period. The base-training period consists of physics-informed learning and data-driven manners. The two-stage fine-tuning period consists of semi-supervised learning and data-driven manners. The above three training manners are **(1)** a methodology grounded in the data-driven manner to use coarse-grained data, **(2)** a data-free physics-informed learning manner using learnable fine-grained state to improve predictive ability, and **(3)** a semi-supervised learning method that capitalizes on the unlabeled data to improve predictive ability further. In both periods, we design the loss function as follows:

$$\mathcal{L} = \beta\mathcal{L}_d + \gamma\mathcal{L}_{ep} + \gamma\mathcal{L}_{tp}, \tag{4}$$

where $\mathcal{L}_d$ is the relative data loss calculated on coarse-grained mesh as: $\mathcal{L}_d = \frac{\|\hat{o}_{t+1} - \tilde{o}_{t+1}\|^2}{\|\tilde{o}_{t+1}\|^2}$ and $\tilde{o}_{t+1}$ is the coarse-grained label, $\beta$ and $\gamma$ are the weights of data loss and physics loss. $\mathcal{L}_{ep}$ and $\mathcal{L}_{tp}$ denote the physics losses of the encoding module and transition module, respectively. By expressing the 4th-order Runge-Kutta (RK4) formulae as $F(\tilde{u}_t, \tilde{u}_{t+1}) = 0$, we design two physics losses $\mathcal{L}_{ep}(\theta) = F(\hat{u}_t(\theta), \hat{u}_{t+1}(\theta))^2$ and $\mathcal{L}_{tp}(\omega) = F(\hat{u}_t, \hat{u}_{t+1}(\omega))^2$. We employ the widely-used standard RK4 method in our framework, and its formulae are listed in the Appendix B.2. In addition, we briefly summarizes the implementation in the Algorithm 1.

**Base-Training Period**: In the base-training period, as we can use the labeled coarse-grained data to achieve the basic predictive performance, we first calculate the relative data loss $\mathcal{L}_d$ training two modules end-to-end in the data-driven manner. Then, for the physics-informed manner, we apply distinct training on each module as follows.

Specifically, for training the encoding module, we train parameters $\theta$ in Eqn. 2 via physics loss to offset the complete lack of fine-grained data. Since only the coarse-grained data is available, we propose to encode the input $\tilde{o}_t$ and label $\tilde{o}_{t+1}$ to learnable fine-grained state $\hat{u}_t$ and $\hat{u}'_{t+1}$ which can be used to calculate the encoding physics loss $\mathcal{L}_{ep}$ by finite dimensional approximations in training. Then, the most likely fine-grained state can be learned to overcome the limitation caused by the data-driven manner. Note that $\tilde{o}_t$ and $\tilde{o}_{t+1}$ share the same encoding module to output the learnable fine-grained states. Moreover, to make the learned $\hat{u}_t$ and $\hat{u}'_{t+1}$ more reliable, we fully utilize existing information by hard encoding $\tilde{o}_t$ and $\tilde{o}_{t+1}$ to the corresponding position.

Besides, for training the transition module, we train parameters $\omega$ in Eqn. 2 using $\mathcal{L}_{tp}$. To collaborate with the data loss introduced earlier, we calculate the derivative of physics loss $\mathcal{L}_{tp}$ to transition module independently. $\mathcal{L}_{tp}$ is computed between the input $\hat{u}_t$ and the predicted subsequent $\hat{u}_{t+1}$ in fine-grained mesh. Moreover, the known BCs can be hard encoded to $\hat{u}_t$ when training the transition module, imposing the prior physics knowledge as described in Rao et al. (2023).

**Two-Stage Fine-Tuning Period**: We leverage unlabeled data to tune both modules after the base-training period, to improve models' predictive ability further. As we mentioned in the base-training, the encoding module has to be trained using both input and label, while the transition module can be trained without label. Thus, we can easily fine-tune the transition module with physics loss based on unlabeled data and then fine-tune the encoding module with data loss using original labeled data, so that we can propagate information of unlabeled data and PDEs from the transition module to the encoding module. The fine-tuning of both modules corresponds to two stages, respectively.

Specifically, at the first fine-tuning stage, we apply unlabeled data to tune the transition module using only $\mathcal{L}_{tp}$ without data loss to make predictions more in line with the PDEs, which is called the physics-tuning stage. However, as the encoding module and transition module are first trained

end-to-end via data loss in the base-training period, the physics-tuned transition module mismatches the base-trained encoding module, leading to deteriorating performance in general.

Then, the second fine-tuning stage (data-tuning stage) applies data loss $\mathcal{L}_d$ to tune the encoding module independently, so that we can propagate unlabeled data and physics information to the encoding module. Note that $\mathcal{L}_d$ is calculated based on the original training set without introducing new labeled data. Also, the data efficiency is significantly improved by the above methods, as the labeled data are applied four times to calculate $\mathcal{L}_d$, $\mathcal{L}_{tp}$, $\mathcal{L}_{ep}$ and tune encoding module in the data-tuning stage. The unlabeled data are also beneficial for learning. Such high data efficiency greatly improves the predictive ability of physical system models utilizing insufficient data.

---

**Algorithm 1** Physics-Informed Coarse-grained data Learning (PICL)

---

**Input:** Data set $\mathcal{D}$ and $\mathcal{B}$, Finite dimensional approximation $\tilde{\mathcal{P}}$, Parameters $\theta$ and $\omega$, Gaps between each fine-tuning period $q$, Steps of two-stage of fine-tuning period $m_1$ and $m_2$.
1: **Initialize** $\theta$, $\omega$ of encoding module $f_\theta$ and transition module $f_\omega$.
2: **while** True **do**
3:     **for** $i = 1$ to $q$ **do**                                       ▷ Base-Training Period
4:         Update $\theta$ and $\omega$ using $\mathcal{L}_{ep} + \mathcal{L}_d$, $\mathcal{L}_{tp} + \mathcal{L}_d$, for each $(\tilde{o}_t, \tilde{o}_{t+1})$ in $\mathcal{D}$.
5:     **for** $i = 1$ to $m_1$ **do**                               ▷ Two-Stage Fine-Tuning Period
6:         Update $\omega$ using $L_{tp}$, for each $\tilde{o}_t^\eta$ in $\mathcal{B}$.
7:     **for** $i = 1$ to $m_2$ **do**
8:         Update $\theta$ using $L_d$, for each $(\tilde{o}_t, \tilde{o}_{t+1})$ in $\mathcal{D}$.

---

## 5 EXPERIMENTS

In this section, we test PICL on several benchmarks. For each one, we first compare the data loss $\mathcal{L}_d$ and reconstruction error $\epsilon$ on test set with four baselines shown in Table 1, where $\epsilon = \frac{\|\hat{u}_t - \tilde{u}_t\|^2}{\|\tilde{u}_t\|^2}$ ($\tilde{u}_t$ only used to calculate metrics $\epsilon$ in inference). Second, we compare the data loss $\mathcal{L}_d$ of multi-step prediction on the test set shown in Fig. 2. Finally, we aim to answer the following primary questions: **(1)** Is PICL sensitive to hyperparameters? **(2)** Does the data quantity impact the performance of PICL? **(3)** Does the data quality impact the performance of PICL? To answer the above questions, we evaluate our method in ablation studies. In addition, we study about output resolution impact on cost, different encoding networks, zero-shot super-resolution, and impact of fine-grained data (when available). Due to space limitations, we show the details of them in Appendix B.

### 5.1 EXPERIMENTS SETUP

Before the description of experiments, we introduce four baselines in this part. **PIDL**: physics-informed deep learning that is based on physics constraints computed by finite difference and does not use training data, applied in Liu & Wang (2021) and Gao et al. (2021a). **FNO** (Li et al., 2020b): a powerful neural operator with FFT-based spectral convolutions. **FNO\***: the same encoding module as PICL is attached ahead of FNO to address our problem better, denoted as FNO\*, where $\gamma = 0$ in loss function 4. **PINO\*** (Li et al., 2021): a hybrid approach incorporating data and physics constraints based on FNO to learn neural operator, but with the same modification as FNO\*, where the physics loss is calculated based on $\tilde{o}_t$ and $\hat{o}_{t+1}$. By attaching the same encoding module in FNO\* and PINO\*, we create a more level playing field to evaluate the performance of our proposed method against the modified baselines, thereby providing a clearer measure of our contributions. In addition, we compare the computational cost between our neural network transition module and numerical solver to demonstrate the value of our transition module in Appendix G

Another setup is the benchmarks: wave equation, linear shallow water equation (LSWE), nonlinear shallow water equation with uneven bottom (NSWE), Burgers equation, and Navier-Stokes equation. We first target the wave equation and LSWE (Rosofsky et al., 2023). Next, we test a more challenging benchmark NSWE with some adjustments based on that in Rosofsky et al. (2023), which is more useful in the applications, as NSWE retains all nonlinear terms, including the uneven bottom. The experiments in Burgers equation, and Navier-Stokes equation are shown in Appendix F. In all benchmarks, the ICs are randomly sampled from the Gaussian Random Fields (GRFs), and models learn to generalize to various ICs.

## 5.2 WAVE EQUATION

We introduce the experiment on wave equation in this part as follows:

$$\frac{\partial^2 u}{\partial t^2} + c^2 \left( \frac{\partial^2 u}{\partial x^2} + \frac{\partial^2 u}{\partial y^2} \right) = 0, \tag{5}$$

where $x, y \in [0, 1), t \in [0, 1]$, $u(x, y, 0) = u_0(x, y)$, $u$ is velocity field, $c$ denotes the speed of wave.

After evaluation, four baselines have worse prediction data loss $\mathcal{L}_d$ in modeling wave equation system than our proposed PICL which can be further improved by fine-tuning with unlabeled data. As shown in Table 1, $\mathcal{L}_d$ of PICL w/o fine-tune has more than 8% improvement after informing the PDEs. After fine-tuning with unlabeled data iteratively, $\mathcal{L}_d$ has about 10% improvement on PICL w/o fine-tune and over 17% improvement on FNO*. Compared with other baselines, $\mathcal{L}_d$ of FNO, PIDL, and PINO* are significantly larger than that of PICL with fine-tune. Because, insufficient data limits the models' predictive ability in the data-driven manner like FNO and FNO*. The physics loss calculated directly on coarse-grained data does not improve the predictive ability of the model and has a negative impact on PIDL and PINO*. In addition, we discover PICL with fine-tune simultaneously has over 10%, 49%, and 51% lower $\epsilon$ than those of baselines, which means PICL can reconstruct the more reliable fine-grained state. We consider it a reason why PICL can learn the superior model of physical systems. The above loss is calculated on the one-step prediction. Then, we evaluate our proposed PICL on the multi-step prediction task. As you can see in Fig. 2, $\mathcal{L}_d$ of PICL always has lower values among all steps. Thus, the wave equation modeling with PICL has superior predictive ability not only on one-step task but also on multi-step task than baselines.

Table 1: Comparison between PICL and four baselines on three benchmarks.

| Benchmarks / Methods | Wave Eqn. | | LSWE | | NSWE | |
|---|---|---|---|---|---|---|
| | $\mathcal{L}_d$ | $\epsilon$ | $\mathcal{L}_d$ | $\epsilon$ | $\mathcal{L}_d$ | $\epsilon$ |
| PIDL | 1.28 | 1.19 | 7.60 | **1.38E-3** | 2.34E-1 | 3.16E-1 |
| FNO | 1.11E-1 | - | 7.92E-2 | - | 1.01E-1 | - |
| FNO* | 3.58E-2 | 2.17 | 4.75E-2 | 6.82E-3 | 6.41E-2 | 1.74 |
| PINO* | 1.01 | 2.09 | 7.60 | 1.63E-3 | 2.32E-1 | 3.18E-1 |
| PICL w/o fine-tune | 2.93E-2 | 1.09 | 2.54E-2 | 1.49E-3 | 3.69E-2 | 2.06E-1 |
| PICL with fine-tune | **2.64E-2** | **1.06** | **2.44E-2** | 1.47E-3 | **3.50E-2** | **2.03E-1** |

## 5.3 LINEAR SHALLOW WATER EQUATION

We introduce the experiment on LSWE in this part as follows:

$$\frac{\partial h}{\partial t} + H \left( \frac{\partial u^x}{\partial x} + \frac{\partial u^y}{\partial y} \right) = 0,$$

$$\frac{\partial u^x}{\partial t} - f u^y = -g \frac{\partial h}{\partial x}, \quad \frac{\partial u^y}{\partial t} + f u^x = -g \frac{\partial h}{\partial y}, \tag{6}$$

where $x, y \in [0, 1)$, $t \in [0, 1]$, $h(x, y, 0) = h_0(x, y)$, $u^x(x, y, 0) = 0$, $u^y(x, y, 0) = 0$, $h$ is the surface height, $u^x$ and $u^y$ are the velocity fields of $x$ and $y$ directions. The mean height is H = 100, and we consider the Coriolis coefficient $f = 1$ and gravitational constant $g = 0.01$.

As shown in Table 1, four baselines see the larger data loss $\mathcal{L}_d$ in modeling LSWE system, while PICL w/o fine-tune exhibits an enhancement of over 46% on FNO*. The fine-tuning with unlabeled data brings an improvement of about 4% on PICL w/o the fine-tune and over 48% on FNO*. Except for FNO*, $\mathcal{L}_d$ of FNO, PIDL, and PINO* are also much larger than that of PICL. It is evident that modeling LSWE with PICL has superior predictive ability compared with baselines, and tuning with unlabeled data further enhances this improvement. Additionally, shown in Table 1, PICL with fine-tune has a relatively reduced $\epsilon$, while FNO, FNO* and PINO* have larger reconstruction errors. Although that of PIDL is slightly lower, it focuses on training the model with physics loss only, ignoring the data constraint for prediction. Thus, PICL can balance the relationship between data

and physics constraints so that superior modeling can be achieved. For the multi-step prediction shown in Fig. 2. PICL always has the lower data loss compared with baselines, and the gap between $\mathcal{L}_d$ of PICL and FNO* is increasing along with the prediction steps. The reason is that the growth of cumulative error may slow down due to the model being constrained to meet the PDEs at each step, thus maintaining higher accuracy in multi-step prediction.

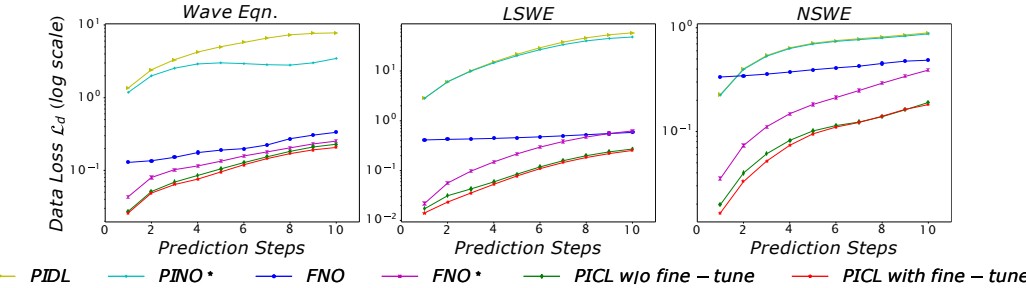

Figure 2: The performance of multi-steps prediction from $1^{st}$ step to $10^{th}$ step on three benchmarks.

## 5.4 NONLINEAR SHALLOW WATER EQUATION WITH UNEVEN BOTTOM

In this section, we evaluate PICL in the more complex NSWE closely to the real world as follows:

$$\frac{\partial(h)}{\partial t} + \frac{\partial(hu^x)}{\partial x} + \frac{\partial(hu^y)}{\partial y} = 0,$$

$$\frac{\partial(hu^x)}{\partial t} + \frac{\partial}{\partial x}[h(u^x)^2 + \frac{1}{2}gh^2] + \frac{\partial(hu^x u^y)}{\partial y} = \nu\left(\frac{\partial^2 u^x}{\partial x^2} + \frac{\partial^2 u^x}{\partial y^2}\right) - fu^y + gh\frac{\partial z}{\partial x}, \quad (7)$$

$$\frac{\partial(hu^y)}{\partial t} + \frac{\partial(hu^x u^y)}{\partial x} + \frac{\partial}{\partial y}[h(u^y)^2 + \frac{1}{2}gh^2] = \nu\left(\frac{\partial^2 u^y}{\partial x^2} + \frac{\partial^2 u^y}{\partial y^2}\right) + fu^x + gh\frac{\partial z}{\partial y},$$

where $x, y \in [0, 1)$, $t \in [0, 1]$, $h(x, y, 0) = h_0(x, y)$, $u^x(x, y, 0) = 0$, $u^y(x, y, 0) = 0$, $z$ is the uneven height of the bottom randomly sampled from GRFs, $\nu$ denotes the viscosity coefficient.

In NSWE system modeling, contrasted with PICL, the baselines exhibit larger $\mathcal{L}_d$. As highlighted in Table 1, PICL w/o fine-tune showcases $\mathcal{L}_d$ with an over 42% improvement compared to FNO*. After fine-tuning with unlabeled data, it is further refined with an uplift of over 5% against PICL w/o fine-tune and over 45% against FNO*. Baselines FNO, PIDL, and PINO* see a similar situation as FNO*. Their $\mathcal{L}_d$ are much larger than that of PICL. The comparison with other baselines is placed in Appendix E. Moreover, we discover that PICL with fine-tune reconstructs the more reliable fine-grained state with $\epsilon$ below 2.03E-1 better than those of baselines. For multi-step prediction in Fig. 2, PICL always performs better than baselines in multiple steps with a similar trend as that in LSWE.

## 5.5 ABLATION STUDIES

**Is PICL sensitive to hyperparameters?** To answer this question, we study the weights of physics loss, lengths of most recent consecutive observations as input, and coefficients in two stages of fine-tuning period on NSWE. We consider weights of physics loss $\gamma \in \{0, 1E-3, 5E-2, 1E-1, 2E-1, 1\}$ in Eqn. 4, lengths of most recent consecutive observations as input $n \in \{1, 2, 4, 6, 8\}$ to the encoding module, and coefficients that include steps in two stages $m_1, m_2 \in \{5, 10, 20\}$ and gaps between each fine-tuning period $q \in \{50, 100, 200, 500\}$. The results of evaluations trained with the above hyperparameters are shown in Fig. 3, and their details are shown in Appendix Table 4.

In Fig. 3(a), we show how varying the weights of each term in the loss function (Eqn. 3) influences the final performance. The results shows that the performance is better when weight $\gamma = $ 1E-1 and 2E-1 than others. On the one hand, a model trained by data-driven ($\gamma = 0$), which only focuses on data (i.e., uses $\mathcal{L}_d$), performs worst. On the other hand, models trained with a significant focus on physics constraints (i.e., $\gamma = 1$) also underperform. In general, PICL with a balance between data and physics loss, leads to superior physical systems modeling.

For lengths of most recent consecutive observations as input shown in Fig. 3(b), $n = 1$ means only the current coarse-grained observation as input, has worse results, and when lengths are longer,

evaluations achieve better results. We consider the reason is that a model input by recent consecutive observations can use temporal features to reconstruct more reliable fine-grained state by physics loss.

For coefficients in the two-stage fine-tuning period shown in Fig. 3(c). When the steps of two stages $m_1 = 10$ and $m_2 = 10$, the evaluation has a good result. This means a model with less $m_1$ and $m_2$ fails to change the performance significantly in the two-stage fine-tuning period. When the gap coefficient $q = 100$, the evaluation has achieved the best result, while a model with less $q$ tunes too frequently, like $q = 50$, causing worse predictive performance than that of $q = 100$.

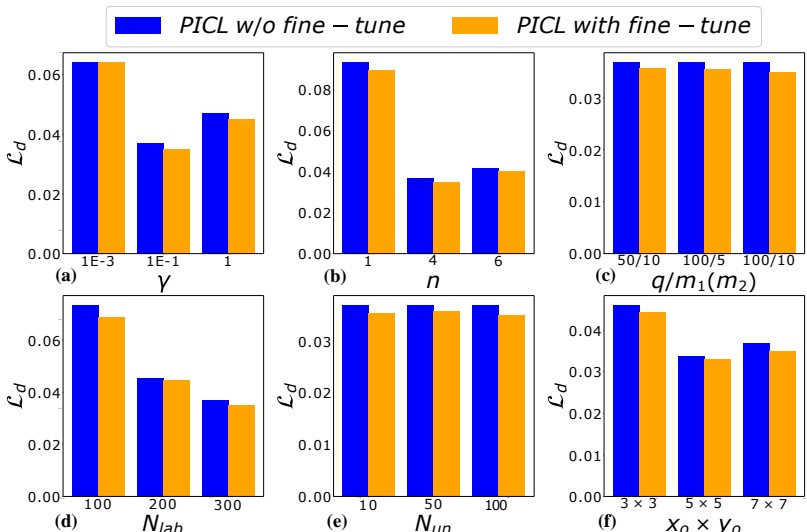

Figure 3: Results of ablation studies. More details of results are presented in Appendix Table 4.

**Does the data quantity impact the performance of PICL?** To evaluate the impact of data quantity for labeled data $\tilde{o}$ and unlabeled data $\tilde{o}^\eta$, we design experiments of $\tilde{o}$ quantity of trajectory $N_{lab} \in \{50, 100, 150, 200, 250, 300, 350\}$ and $\tilde{o}^\eta$ quantity of trajectory $N_{un} \in \{10, 50, 100, 150\}$ on NSWE. The values of evaluations are shown in Fig. 3(d) and 3(e). **(1)** For quantity of labeled data, $\mathcal{L}_d$ decreases along the rising of $N_{lab}$. More quantity of labeled data leads to better predictive ability of models. **(2)** For quantity of unlabeled data, we can see no matter how many $N_{un}$ is, $\mathcal{L}_d$ can be reduced by two-stage fine-tuning using unlabeled data. Moreover, among the range of $N_{un}$ and $N_{lab}$, fine-tuning can always bring improvements for PICL by introducing unlabeled data.

**Does the data quality impact the performance of PICL?** To answer the question, we study the quality of input and output data with the different coarse-grained sizes $x_o \times y_o \in \{3 \times 3, 5 \times 5, 7 \times 7, 11 \times 11\}$, based on NSWE. As smaller sizes have less spatial information, we would consider this feature as the lower quality. The results of the evaluations are shown in Fig. 3(f). Evaluation is worse in smaller coarse-grained sizes like $3 \times 3$, and those are better when coarse-grained sizes are relatively larger. When the coarse-grained size is small with low data quality, the encoding module faces the challenge of reconstructing the fine-grained state with physics loss using less spatial information, and the larger coarse-grained size has relatively more spatial information. Furthermore, among the range of $x_o \times y_o$, PICL with fine-tune can always bring improvements.

## 6 CONCLUSION

In this paper, we presented PICL, a physics-informed coarse-grained data learning framework for enhanced modeling of physical systems under limited coarse-grained data conditions. Within PICL, we employ the encoding module and transition module in tandem, and devise a training strategy with two periods to address challenges associated with coarse-grained data quality and insufficient data quantity. Using Wave Eqn., LSWE, and NSWE as examples, we demonstrated that PICL can improve prediction accuracy and data efficiency, as well as reconstruct more reliable fine-grained states without the need for fine-grained data.

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

## A  MODEL ARCHITECTURE

Here, we detail the architecture of model with PICL, complementary to Sec. 4.1. This architecture is used throughout all experiments, with only adjusting a few hyperparameters (e.g., input dimension, latent dimension) depending different settings. We detail the encoding module $f_\theta$ and transition module $f_\omega$, and the architectures used in the Wave Eqn., LSWE, and NSWE experiments. A summary of the hyperparameters is also provided in Table 2.

**Wave Eqn.**: For the experiments in wave equation, including models of PICL w/o fine-tune and PICL with fine-tune, our encoding module $f_\theta$ employs a U-Net with the number of 23 residual blocks. The input dimension is $(2n, 9, 9)$, and the output dimension of the encoding module is $(2, 41, 41)$, which is the fine-grained state. The transition module $f_\omega$ employs an FNO model with the number of 4 FNO layers and a layer width of 32. It uses the GeLu activation (Li et al., 2020b). The input dimension is $(2, 41, 41)$, and that of the output is $(2, 41, 41)$. The down-sampling block employs a gap of 5 to get the predicted subsequent coarse-grained observation that has $(2, 9, 9)$ dimension. For both coarse-grained observation and fine-grained state, two channels of the first dimension represent the $u$ that is the field of velocity in Eqn. 5, and $\phi$ that is the field of velocity potential quite related to $u$ in wave equation. The second and third dimensions are the spatial dimensions, which are 9 for coarse-grained observation and 41 for fine-grained state. In this setting, the coarse proportion is less than 4.82%.

**LSWE and NSWE**: For the experiments in shallow water equation, including models of PICL w/o fine-tune and PICL with fine-tune, our encoding module $f_\theta$ employs a U-Net with the number of 23 residual blocks. The input dimension is $(3n, 7, 7)$, and the output dimension of the encoding module is $(3, 32, 32)$, which is the fine-grained state. The transition module $f_\omega$ employs an FNO model with a number of 4 FNO layers, and the width is 32. It uses the GeLu activation (Li et al., 2020b). The input dimension is $(3, 32, 32)$, and that of the output is $(3, 32, 32)$. The down-sampling block employs a gap of 5 to get the predicted subsequent coarse-grained observation that has $(3, 7, 7)$ dimension. For both coarse-grained observation and fine-grained state, three channels of the first dimension represent the $u^x$ and $u^y$ that are the fields of velocity in $x$ and $y$ directions, and $h$ that is the field of fluid column height, in Eqn. 6 and 7. The second and third dimensions are the spatial dimensions, which are 7 for coarse-grained observation and 32 for fine-grained state. In this setting, the coarse proportion is less than 4.79%.

Table 2: Hyperparameters used for model architecture.

| Hyperparameters name for model architecture | Wave Eqn. | LSWE | NSWE |
|---|---|---|---|
| $f_\theta$: Input dimension | $(2, 9, 9)$ | $(3, 7, 7)$ | $(3, 7, 7)$ |
| $f_\theta$: Output dimension | $(2, 41, 41)$ | $(3, 32, 32)$ | $(3, 32, 32)$ |
| $f_\theta$: Residual block number | 23 | 23 | 23 |
| $f_\theta$: Channel | 32 | 32 | 32 |
| $f_\theta$: Dropout | 0.1 | 0.1 | 0.1 |
| $f_\theta$: Activation function | Swish | Swish | Swish |
| $f_\omega$: Input dimension | $(2, 41, 41)$ | $(3, 32, 32)$ | $(3, 32, 32)$ |
| $f_\omega$: Output dimension | $(2, 41, 41)$ | $(3, 32, 32)$ | $(3, 32, 32)$ |
| $f_\omega$: Layers number | 4 | 4 | 4 |
| $f_\omega$: Modes | 12 | 12 | 12 |
| $f_\omega$: Width | 32 | 32 | 32 |
| $f_\omega$: Activation function | GeLu | GeLu | GeLu |
| Down-sampling gap | 5 | 5 | 5 |
| Down-sampling output dimension | $(2, 9, 9)$ | $(3, 7, 7)$ | $(3, 7, 7)$ |

## B  IMPLEMENTATION DETAILS

In this section, we provide experiment details for three benchmarks. First, we introduce the details of data generation by FDM. Then, we introduce the manner of physics loss calculation during the training. After that, the details about base-training and two-stage fine-tuning periods are supplemented. Table 3 shows general hyperparameters of training, except for the hyperparameters that are

modified in ablation studies introduced in Sec. 5.5. Finally, we detail the hard encoding of known physics information.

---

**Algorithm 2** PICL (a detailed version)

---

**Input:** Labeled data set $\mathcal{D}$, Unlabeled data set $\mathcal{B}$, Finite dimensional approximations $\tilde{\mathcal{P}}$, Trained parameters $\theta$ and $\omega$ of encoding module $f_\theta$ and transition module $f_\omega$, Gaps between each fine-tuning period $q$, Steps of two-stage of fine-tuning period $m_1$ and $m_2$.
1: **Initialize** parameters $\theta$ and $\omega$.
2: **while** True **do**
3:     **for** $i = 1$ to $q$ **do**                                                   ▷ Base-Training Period
4:         **for** each $(\tilde{o}_t, \tilde{o}_{t+1})$ in $\mathcal{D}$ **do**
5:             Encode $\{\tilde{o}_{t-i}\}_{i=0}^n$ to $\hat{u}_t$ using $f_\theta$.
6:             Encode $\{\tilde{o}_{t+1-i}\}_{i=0}^n$ to $\hat{u}'_{t+1}$ using $f_\theta$.
7:             Calculate $\mathcal{L}_{ep}$ based on $\tilde{\mathcal{P}}$.
8:             Predict $\hat{u}_{t+1}$ using $f_\omega$.
9:             Calculate $\mathcal{L}_{tp}$ based on $\tilde{\mathcal{P}}$.
10:            Down-sample $\hat{o}_{t+1}$.
11:            Calculate data loss $\mathcal{L}_d$.
12:            Update $\theta$ and $\omega$ with joint loss $\mathcal{L}_{ep} + \mathcal{L}_d$ and $\mathcal{L}_{tp} + \mathcal{L}_d$.
13:     **for** $i = 1$ to $m_1$ **do**                                ▷ Two-Stage Fine-Tuning Period
14:         **for** each $\tilde{o}_t^\eta$ in $\mathcal{B}$ **do**
15:             Encode $\{\tilde{o}_{t-i}\}_{i=0}^n$ to $\hat{u}_t$ using $f_\theta$.
16:             Predict $\hat{u}_{t+1}$ using $f_\omega$.
17:             Calculate transition physics loss $L_{tp}$ based on $\tilde{\mathcal{P}}$.
18:             Update $\omega$ using loss $L_{tp}$.
19:     **for** $i = 1$ to $m_2$ **do**
20:         **for** each $(\tilde{o}_t, \tilde{o}_{t+1})$ in $\mathcal{D}$ **do**
21:             Encode $\{\tilde{o}_{t-i}\}_{i=0}^n$ to $\hat{u}_t$ using $f_\theta$.
22:             Predict $\hat{u}_{t+1}$ using $f_\omega$.
23:             Down-sample $\hat{o}_{t+1}$.
24:             Calculate data loss $L_d$.
25:             Update $\theta$ using loss $L_d$.

---

## B.1 DATA GENERATION

To generate the training data, we initiate ICs randomly sample from the GRFs and subsequently evolved them both spatially and temporally. Specifically, we employed the RK4 method for temporal evolution, starting from $t = 0$ and progressing up to $t = 1$ with a time-step of $\delta t = 0.01$ in the same fashion to Rosofsky et al. (2023). For the computation of spatial derivatives, a fourth-order central difference scheme was utilized within the framework of FDM. As we aim to generalize the model on a variety of ICs, we divide the training data set $\mathcal{D}$, and test data set $\mathcal{D}'$ and unlabeled data set $\mathcal{B}$ based on trajectories generated from i.i.d. ICs. For ease of reading, in this paper, we illustrate the PICL for PDE-governed physical systems in the 2-D examples with regular mesh and finite difference approximation.

## B.2 CALCULATION OF PHYSICS LOSS

Physics loss both $\mathcal{L}_{ep}$ and $\mathcal{L}_{tp}$ are calculated as similar method like data generation. By discretizing the PDEs with fourth-order central-difference scheme and the RK4 time discretization method on fine-grained mesh ($41{\times}41$ in Wave Eqn., $32{\times}32$ in LSWE and NSWE), the RK4 to solve differential

equation $u' = f(x, y, u)$ can be expressed as:

$$
\begin{aligned}
k_1 &= f(x, y, u_t), \\
k_2 &= f\left(x + \frac{h}{2}, y + \frac{h}{2}, u_t + \frac{h}{2}k_1\right), \\
k_3 &= f\left(x + \frac{h}{2}, y + \frac{h}{2}, u_t + \frac{h}{2}k_2\right), \\
k_4 &= f\left(x + h, y + h, u_t + hk_3\right), \\
u_{t+1} &= u_t + \frac{h}{6}(k_1 + 2k_2 + 2k_3 + k_4).
\end{aligned}
\tag{8}
$$

Let a function $F$ denotes the RK4 calculation, where $F(u_t, u_{t+1}) = 0$. We design $F^2$ as the physics loss. Specifically, to the encoding physics loss:

$$
\mathcal{L}_{ep}(\theta) = F(u_t(\theta), u_{t+1}(\theta))^2.
\tag{9}
$$

To the transition physics loss:

$$
\mathcal{L}_{tp}(\omega) = F(u_t, u_{t+1}(\omega))^2.
\tag{10}
$$

### B.3 LEARNING STRATEGY

**Base-Training Period**: In the base-training period, we jointly train the parameters $\theta$ and $\omega$ in Eqn. 2 and 3 for both the encoding module and the transition module. This is done using a combination of data-driven and physics-informed methods. A significant challenge we face is the absence of fine-grained data, as we can only access the coarse-grained data $\tilde{o}_t$ and $\tilde{o}_{t+1}$. To solve this challenge, we introduce physics loss terms $\mathcal{L}_{ep}$ and $\mathcal{L}_{tp}$ for the encoding and transition modules, respectively. The encoding module is trained to map the coarse-grained inputs $\tilde{o}_t$ and $\tilde{o}_{t+1}$ to learnable fine-grained states $\hat{u}_t$ and $\hat{u}'_{t+1}$, which can then be used to compute the $\mathcal{L}_{ep}$ using manner in last paragraph. In a similar way, the transition module is trained using $\mathcal{L}_{tp}$. By collaboratively training these modules via $\mathcal{L}_d$, $\mathcal{L}_{ep}$ and $\mathcal{L}_{tp}$, we are able to reconstruct the more reliable and reasonable fine-grained state without requiring fine-grained data and improve the predictive ability of physical system models, thereby overcoming the limitations of data-driven supervised approaches.

**Two-Stage Fine-Tuning Period**: In two-stage fine-tuning period, given the abundance of readily available unlabeled, coarse-grained data for PDE-governed physical systems, we first focus on the transition module in what we term the physics-tuning stage. In this stage, the module is fine-tuned independently using the $\mathcal{L}_{tp}$ calculated using the unlabeled input (learnable fine-grained state) and the predicted subsequent fine-grained state, thereby enhancing its prediction to better align with PDEs. However, this physics-tuned transition module is not inherently compatible with the encoding module, which was trained during the base-training period. This incongruence results in a gradual deterioration in performance. To address this problem, a second fine-tuning stage, termed the data-tuning stage, is introduced. Here, the encoding module is fine-tuned individually using the data loss $\mathcal{L}_d$, which is computed based on the original training set $\mathcal{D}$ without the introduction of new labeled data. By proceeding in this manner, we effectively propagate the information of PDEs and unlabeled data from the transition module into the encoding module. The pipline of fine-tuning is illustrated in Fig. 4. The detailed implementation summary is in Algorithm 2.

### B.4 HARD ENCODING OF PHYSICS INFORMATION

While incorporating physics loss serves as a soft constraint during training, it is essential to encode hard constraints based on known physics information, such as BCs, coarse-grained observations, and sampled data positions. Super-resolution of coarse-grained resolutions is inherently an ill-posed problem due to the daunting task of extrapolating from limited data. However, in scientific tasks, prior knowledge—such as governing equations and BCs—is often available and can offer invaluable guidance. In this regard, we hard encode the known physics information, specifically BCs, directly into the model to facilitate training in the same fashion to Rao et al. (2023). Given the uniform discretization of spatial domains, encoding BCs is conveniently achieved through pixel-wise padding techniques.

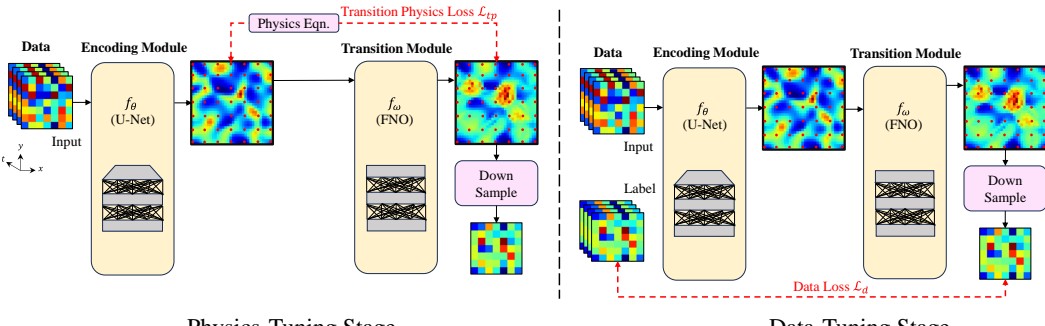

Figure 4: Two-Stage Fine-Tuning Period. Physics-tuning stage (left) and Data-tuning stage (right).

1. For Dirichlet BCs, we can explicitly incorporate the known boundary values into the fine-grained state.
2. For Neumann BCs, we can introduce ghost nodes to infer external node values using finite difference methods. This allows for a time-invariant physical relationship between ghost nodes and internal nodes.
3. For periodic BCs, we can employ periodic padding techniques to encode spatial continuity. This approach reuses boundary values from the opposite side of the mesh.

Through these hard constraints, the model is provided with accurate descriptions of underlying dynamics at the boundary and effective guidance for reconstruction tasks, thereby enhancing both the training process and the model's predictive ability.

The task of encoding coarse-grained observations into fine-grained size is complex, particularly when these observations span multiple data dimensions such as velocities, pressures, and spatial positions. These coarse-grained data points serve as preliminary knowledge for learning the hidden, fine-grained state. When the coarse-grained observation is encoded to the learnable fine-grained state, coarse-grained observation can be hard encoded to the sampled position by the encoding module, so that the learned state can be more reliable. On the other hand, as the position information is the preliminary information, the down-sample can filter from the predicted subsequent fine-grained state to the predicted subsequent coarse-grained observation. By doing so, we do not need to use a neural network to fit the subsequent coarse-grained observation, which avoids introducing more parameters. This approach leads to a significant improvement in the data efficiency and predictive ability of the model with PICL.

Table 3: Hyperparameters used for training.

| Hyperparameters name for training | Value |
|---|---|
| Batch size | 32 |
| $\alpha$, Learning rate | 1 |
| $\lambda$: Weight decay | 1E-4 |
| $\delta t$: Time gap | 0.01 |
| $\gamma$: Scheduler factor | 0.5 |
| $\beta$: Weight of data loss | 1E-3 |
| Training epochs | 1000 |

## B.5 DETAILS OF DOWN-SAMPLING OPERATION

For a given fine-grained state $\tilde{u}$, represented as an $n \times n$ matrix where $\tilde{u}(x, y)$ denotes the value at coordinates $(x, y)$, we define a sampling set $\Phi$ as follows:

$$\Phi = \{(x, y) \mid x, y \in \{0, k, 2k, \ldots, mk\}\},$$

where $m$ is the largest integer less than or equal to $(n-1)/k$ and a multiple of $k$, signifying that $\Phi$ includes the coordinates of every $k$-th value in the $\tilde{u}(x, y)$. Applying the down-sampling operation,

the resultant coarse-grained points $P$ is obtained:

$$P = \{\tilde{u}(x,y) \mid (x,y) \in \Phi\},$$

Here, $P$ is a set of values obtained from the fine-grained state $\tilde{u}(x,y)$ according to the sampling positions in set $\Phi$. Rearrange the set $P$ to the $r \times r$ matrix of coarse-grained observation $\tilde{o}(x_o, y_o)$, where $(x_o, y_o) \in \{0, 1, 2, \ldots, m\}$. The training coarse-grained data set $\mathcal{D} = \{\tilde{o}_i\}_{i=0}^{N}$ is down-sampled from the fine-grained state.

Moreover, as the coordinates set $\Phi$ of coarse-grained data is available, we apply a type of hard encoding method to utilize the position information. Thus, the output of the model is down-sampled on the predicted subsequent fine-grained state from $\Phi$, instead of using a neural network to avoid introducing more trainable parameters.

### B.6 DETAILS OF ABLATION STUDIES

**Is PICL sensitive to hyperparameters?** We evaluate the sensitive of the weights given $\mathcal{L}_{tp}$ and $\mathcal{L}_{ep}$, the input covering lengths of most recent consecutive observations, and the coefficients in the two-stage fine-tuning period on the performance of PICL for NSWE setting. As the results of FNO* are relatively better than other baselines, serving as a representative for them, we focus on the improvement between FNO* and PICL in the following ablation studies.

For weights of physics loss, as presented in Eqn. 4, there are $\mathcal{L}_d$, $\mathcal{L}_{tp}$ and $\mathcal{L}_{ep}$, where $\mathcal{L}_{tp}$ and $\mathcal{L}_{ep}$ are weighted with coefficient $\gamma$. Accordingly, we study the influence of the hyperparameter $\gamma$ in the loss function given in Eqn. 4 on the predictive accuracy. For this purpose, we consider $\gamma \in \{0, 1\text{E-}3, 5\text{E-}2, 1\text{E-}1, 2\text{E-}1, 1\}$. The values of the evaluations for PICL based on NSWE trained with each of the $\gamma$ values are presented in Table 4. $\gamma = 0$ indicates the FNO* depends on the $\mathcal{L}_d$, and $\mathcal{L}_{tp}$ and $\mathcal{L}_{ep}$ are not accounted for. As presented in Table 4, the best performance on the test set is achieved for a loss function with weighting coefficients of $\gamma = 1\text{E-}1, 2\text{E-}1$. Allover this work, we refer to the weight coefficient with $\gamma = 1\text{E-}1$. A model that is trained by data-driven, which only focuses on the data (i.e., uses $\mathcal{L}_d$ only) and does not account for the physics constraints (i.e., ignores $\mathcal{L}_{tp}$ and $\mathcal{L}_{ep}$) underperform than that of PICL. On the other hand, models trained with a significant focus on the physics constraints only (i.e., large $\gamma$) also underperform. In general, a balance between the focus of the PICL on the data and the physics constraints leads to an optimal performance.

For lengths of most recent consecutive observations as input, as introduced in Sec. 4.2, there are most recent consecutive observations input to the model, balanced with length coefficient $n$, fed to the encoding module to make up for the lack of spatial information with the quantity of temporal information, inspired by calculation of FDM. Accordingly, we study the influence of the hyperparameter $n$, deciding the temporal features carried by input data on the performance of PICL. For this purpose, we consider $n \in \{1, 2, 4, 6, 8\}$. The values of the evaluations for PICL based on NSWE trained with each of the $n$ values are presented in Table 4. $n = 1$ indicates the input only has the current coarse-grained observation, and the most recent consecutive observations are not accounted for. As presented in Table 4, when covering the coefficient of $n = 4$, the evaluation on the test set has achieved a good result. A model that is trained by input without the most recent consecutive observations faces a challenge when the encoding module reconstruct fine-grained state leveraging insufficient spatial information by physics loss. Allover this work, we refer to this length coefficient $n = 4$.

For the coefficients in the two-stage fine-tuning period, in the physics-tuning stage, unlabeled data are applied to tune the transition module using only $\mathcal{L}_{tp}$ without $\mathcal{L}_d$. Then, we apply $\mathcal{L}_d$ to tune the encoding module independently in the data-tuning stage, where $\mathcal{L}_d$ is calculated based on the original training set without introducing new labeled data. The steps of the two stages are controlled by the coefficients $m_1$ and $m_2$ to make PICL achieve better performance in evaluation. Accordingly, we study the impact of the hyperparameters $m_1$ and $m_2$. For this purpose, we consider $m_1 \in \{5, 10, 20\}$ and $m_2 \in \{5, 10, 20\}$. The values of the evaluations for PICL based on NSWE trained with each of the $m_1$ and $m_2$ values are presented in Table 4. As presented in Table 4, when the coefficients $m_1 = 10$ and $m_2 = 10$, the evaluation on the test set has achieved a good result. A model with less $m_1$ and $m_2$ fails to modify the performance significantly, while a model with more steps causes too much deterioration in the physics-tuning stage. Allover this work, we refer to this optimum coefficients $m_1 = 10$ and $m_2 = 10$. Another coefficient is the gap between each two-stage fine-tuning period. We further study the impact of gap coefficient $q$. For this purpose, we consider

Table 4: Results of ablation studies. Evaluation of **Impact on Hyperparameters**, **Impact of Data Quantity** and **Impact of Data Quality**. In weight of phys. loss, $\gamma = 0$ is a model trained by data loss only, and both rows have the same values as such model cannot be tuned with physics loss. In coefficients in fine-tuning period and quantity of unlabeled data, there are only results with fine-tune as these ablation studies are not related to PICL w/o fine-tune, which briefly plotted in Fig. 3 using the same value of PICL w/o fine-tune in Table 1. In quality of data, $3*, 5*, 7*$ and $11*$ denote the coarse-grained sizes $3 \times 3$, $5 \times 5$, $7 \times 7$, $11 \times 11$.

| | $\gamma$ | 0 | 1E-3 | 5E-2 | 1E-1 | 2E-1 | 1 | - |
|---|---|---|---|---|---|---|---|---|
| Weight of Phys. Loss | PICL w/o fine-tune | 6.41E-2 | 4.40E-2 | 3.91E-2 | 3.69E-2 | **3.61E-2** | 4.70E-2 | - |
| | PICL with fine-tune | 6.41E-2 | 4.16E-2 | 3.61E-2 | **3.50E-2** | 3.52E-2 | 4.49E-2 | - |
| | $n$ | 1 | 2 | 4 | 6 | 8 | - | - |
| Length of Most Recent Consecutive Observations | PICL w/o fine-tune | 9.32E-2 | **3.50E-2** | 3.69E-2 | 4.15E-2 | 5.09E-2 | - | - |
| | PICL with fine-tune | 8.94E-2 | 3.51E-2 | **3.50E-2** | 4.00E-2 | 4.94E-2 | - | - |
| | $q$ | 50 | 100 | 100 | 100 | 100 | 200 | 500 |
| | $m_1$ | 10 | 5 | 10 | 10 | 20 | 10 | 10 |
| Coefficients in Fine-Tuning Period | $m_2$ | 10 | 5 | 10 | 5 | 20 | 10 | 10 |
| | PICL with fine-tune | 3.58E-2 | 3.55E-2 | **3.50E-2** | 3.53E-2 | 6.93E-2 | 3.58E-2 | 3.64E-2 |
| | $N_{lab}$ | 50 | 100 | 150 | 200 | 250 | 300 | 350 |
| Quantity of Labeled Data | PICL w/o fine-tune | 9.53E-2 | 7.34E-2 | 5.61E-2 | 4.53E-2 | 3.97E-2 | 3.69E-2 | **3.32E-2** |
| | PICL with fine-tune | 8.81E-2 | 6.88E-2 | 5.39E-2 | 4.48E-2 | 4.01E-2 | 3.50E-2 | **3.21E-2** |
| | $N_{un}$ | 10 | 50 | 100 | 150 | - | - | - |
| Quantity of Unlabeled Data | PICL with fine-tune | 3.53E-2 | 3.58E-2 | **3.50E-2** | 3.56E-2 | - | - | - |
| | $x_o \times y_o$ | 3* | 5* | 7* | 11* | - | - | - |
| Quality of Data | PICL w/o fine-tune | 4.60E-2 | **3.38E-2** | 3.69E-2 | 3.90E-2 | - | - | - |
| | PICL with fine-tune | 4.43E-2 | **3.30E-2** | 3.50E-2 | 3.84E-2 | - | - | - |

$q \in \{50, 100, 200, 500\}$ and control the same training epoch of the base-training period. The values of the evaluations for PICL on NSWE trained with each of the $q$ values are presented in Table 4. When the gap coefficient $q = 100$, the evaluation of the test set has achieved the best result. A model with less $q$ tunes too frequently, causing instability of base-training, while a model with more $q$ does not have an obvious impact on results. Allover this work, we refer to this optimum gap coefficient $q = 100$.

**Does the data quantity impact the performance of PICL?** In this part, we evaluate the impact of the quantity of labeled data $\tilde{o}$ and unlabeled data $\tilde{o}^{\eta}$ when the physics information is integrated on the performance of PICL for the NSWE setting. As presented in Sec. 4.2, $\mathcal{L}_d$, $\mathcal{L}_{tp}$ and $\mathcal{L}_{ep}$ are applied to train the neural network collaboratively, where $\mathcal{L}_d$ is calculated by $\tilde{o}$. Note that we can only access $\tilde{o}$ in the real world generally. It is interesting to evaluate the performance of PICL on different quantities of $\tilde{o}$. Moreover, we assume some $\tilde{o}^{\eta}$ is accessible for two-stage fine-tuning. The quantity of $\tilde{o}^{\eta}$ also impacts the result of PICL.

For quantity of $\tilde{o}$, we study the influence of the $\tilde{o}$ quantity $N_{lab}$ on the predictive accuracy. For this purpose, we consider $N_{lab} \in \{50, 100, 150, 200, 250, 300, 350\}$. The values of the evaluations for PICL on NSWE trained with each of the $N_{lab}$ values are presented in Table 4. As we aim to generalize the model on a variety of ICs, data number $N_{lab}$ denotes the number of trajectories beginning from different ICs. The $\mathcal{L}_d$ decreases along with the increasing of $N_{lab}$. In the range of $N_{lab}$, PICL can always bring improvements based on results in Table 4.

For quantity of $\tilde{o}^{\eta}$, we study the influence of the $\tilde{o}^{\eta}$ quantity $N_{un}$ on the predictive accuracy. For this purpose, we consider $N_{un} \in \{10, 50, 100, 150\}$. The values of the evaluations for PICL on NSWE trained with each of the $N_{un}$ values are presented in Table 4. The $\tilde{o}^{\eta}$ number $N_{un}$ denotes the number of trajectories beginning from different ICs with labels during training. We can see no matter how much $N_{un}$ is, $\mathcal{L}_d$ can be reduced by two-stage fine-tuning. On the other hand, the FNO and FNO* cannot apply such $\tilde{o}^{\eta}$. In the range of $N_{un}$, fine-tuning can always bring improvements for PICL by introducing the unlabeled data compared with baselines.

**Does the data quality impact the performance of PICL?** In this part, we evaluate the quality of input and output data with the different coarse-grained sizes based on NSWE. In the real world, the measured network not only uses a single size but also uses different sizes of measurement according to actual situations. As smaller sizes have less spatial information, we would consider this feature as the lower quality. We generate four data sets with sizes $x_o \times y_o \in \{3 \times 3, 5 \times 5, 7 \times 7, 11 \times 11\}$. The values of the evaluations for PICL trained with each size of the $x_o \times y_o$ values are presented in Table 4. $\mathcal{L}_d$ calculated between prediction and ground truth are worse in small coarse-grained sizes like $3 \times 3$ as the encoding module faces the challenge of reconstructing the fine-grained state only using less spatial knowledge and physics loss. It has better results when coarse-grained sizes have relatively much spatial knowledge. Moreover, In the range of $x_o \times y_o$, PICL with fine-tune can always bring improvements based on results in Table 4.

**When the output resolution of encoder module increases, how much does it impact the computational cost?** As the computation of physics loss leads to more computational cost than the data-driven manner, we study the increasing of computational cost when the output resolution of the encoder module $f_{\theta}$ increases based on the NSWE setting. We employ the hyperparameters in the same way as the experiment in Sec. 5.4 shown in Table 2 and Table 3, and calculate the computational time of the model's inference and batch training. We compare the computational time of the proposed PICL with baseline FNO* to illustrate the variance between physics-informed training manner and data-driven manner.

Table 5: Inference Computational Time of Different Output Resolutions of $f_{\theta}$.

| Method | $32 \times 32$ (s) | $48 \times 48$ (s) | $64 \times 64$ (s) |
|--------|--------|--------|--------|
| FNO* | 0.0751 | 0.1445 | 0.2387 |
| PICL | 0.0764 | 0.1398 | 0.2374 |

From the Table 5, we can see no matter how much the output resolution of $f_{\theta}$ is, the time of inference of FNO* and PICL always have a similar cost as they have the same model structure. In Table 6, for the time of training, the cost of PICL is larger than that of FNO* as they have different training

Table 6: Training Computational Time of Different Output Resolutions of $f_\theta$.

| Method | $32 \times 32$ (s) | $48 \times 48$ (s) | $64 \times 64$ (s) |
|--------|-------------|-------------|-------------|
| FNO*   | 0.4555      | 1.5786      | 2.8061      |
| PICL   | 0.8007      | 2.5895      | 3.8649      |

loss functions, and PICL is required to calculate the physics losses and auto-differentiate on two modules, but the time cost is acceptable.

**Does the encoding module can be replaced by another architecture?** The aim of this work is to propose a learning framework that can apply physics information on coarse-grained data to improve predictive ability, whether in numerical simulations or real-world applications. In practical implementation, the encoding process is similar to the image processing task, which aims to extract features from coarse-grained input. In response to this, we follow the works and employ the U-Net architecture (Ronneberger et al., 2015) as the encoding module, a proven model well-suited for tasks akin to super-resolution (Esmaeilzadeh et al., 2020). It is only a choice guided by the specific requirements of our study. Future work could explore the efficacy of alternative neural network architectures for these modules. In this paper, we replace the U-Net with the Transformer to answer the question, based on the NSWE setting. We employ the official implementation of the Transformer in ViT (Dosovitskiy et al., 2020), which is a deep learning architecture for computer vision tasks that leverages the Transformer model's self-attention mechanism to process images as sequences of tokens. The results are shown in Table 7.

Table 7: $\mathcal{L}_d$ of model with U-Net and Transformer as encoding module.

| Method | FNO* | PICL w/o fine-tune | PICL with fine-tune |
|--------|------|--------------------|---------------------|
| U-Net  | **6.41E-2** | **3.69E-2** | **3.50E-2** |
| Transformer | 6.79E-2 | 5.01E-2 | 4.70E-2 |

We can see that PICL can still have the improvement compared with the data-driven manner FNO* by incorporating the physics information, especially when it is fine-tuned using the unlabeled coarse-grained data after replacing the encoding module from U-Net to Transformer. However, the $\mathcal{L}_d$ of the Transformer encoding module is slightly larger than that of the U-Net encoding module whether the training manner is.

**How to address the zero-shot super-resolution cases?** We study the zero-shot super-resolution generalization ability of the proposed PICL to new coarse-grained meshes based on the NSWE setting. The original U-Net encoding module requires consistent coarse-grained meshes between inference and training, making it difficult to complete the zero-shot super-resolution cases. The goal of our work is to develop a training framework that is not limited to the specific encoding module. In this experiment, we replace the U-Net with a Transformer as an encoding module and release the zero-shot super-resolution case following the method in MAgNet (Boussif et al., 2022). MAgNet is a mesh-based neural operator that enables zero-shot generalization to new non-uniform meshes and long-term prediction. We apply the nearest neighbors interpolation as same as MAgNet in the encoding module so that it can address the zero-shot super-resolution cases. We train the model on $7 \times 7$ coarse-grained mesh, and test zero-shot super-resolution on $3 \times 3, 5 \times 5, 7 \times 7, 11 \times 11$. We use the reconstruction error $\epsilon$ to measure the zero-shot super-resolution performance. The results are plotted in Fig. 5.

The data presented in the Fig. 5 shows that the PICL with fine-tune outperforms the FNO* method, whether the inference size of the coarse-grained mesh is. It indicates that the incorporation of physics information can significantly improve the zero-shot super-resolution of the encoding module, by ensuring that the output adheres to the underlying physics equation, especially when fine-grained data is not available.

**Can fine-grained data (when available) improve the performance of PICL?** Although the setting of the proposed PICL is the scenario where only coarse-grained data is available, it is still interesting to explore if the fine-grained data (when available) can help the training and improve the

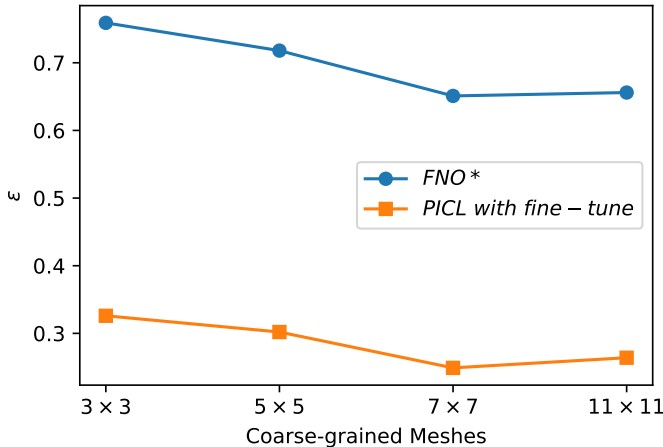

Figure 5: Zero-shot super-resolution performance on NSWE setting

performance of the proposed PICL. We evaluate this situation using the experiment on NSWE and assume the coarse-grained observations have the fine-grained data.

We consider three cases when we assume the fine-grained data is available, including 1/3, 1/2, and all coarse-grained data have the corresponding fine-grained data. Then, we apply the fine-grained data as the labels of the encoding module so that it can learn to reconstruct the more accurate and reliable fine-grained state. As the encoding module with fine-grained labels does not need fine-tuning, we do the experiment using the PICL w/o fine-tune. From Table 8, we can see that both the baseline FNO* (without physics loss) and our proposed PICL benefit from the increasing of fine-grained data. It is worth mentioning that the improvements are over 38% and 12% when all the fine-grained data are assumed to be available for FNO* and PICL, respectively. Moreover, these results also highlight an important aspect of our framework: the ability to use the known physics information as a substitute for fine-grained data in training the encoding module, especially in scenarios where such data is unavailable. The smaller improvement observed in PICL compared to the FNO*, when fine-grained data is available, suggests that PICL can effectively compensate for the absence of fine-grained data to learn the state.

Table 8: $\mathcal{L}_d$ of model with or without fine-grained data.

|  | FNO* | PICL |
|---|---|---|
| w/o fine-grained data | 6.41E-2 | 3.69E-2 |
| 1/3 with fine-grained data | 5.13E-2 | 3.41E-2 |
| 1/2 with fine-grained data | 5.01E-2 | 3.35E-2 |
| all with fine-grained data | **4.70E-2** | **3.29E-2** |

## C  BASELINES

Here, we provide additional details on the baselines used in experiments. The meanings of baselines are introduced in Sec. 5.1. In the following, we mainly give details on their implementation.

**PIDL**: A physics-informed deep learning methodology that is purely based on soft physics constrains computed by finite difference and do not use training data which is utilized in a model-based reinforcement learning (Liu & Wang, 2021). The calculation method of physics loss is the same as PhyGeoNet (Gao et al., 2021a). As we aim to compare the performance between PICL and the model purely trained by physics loss, PIDL has the same architecture as PICL except for training via physics loss independently calculated as Liu & Wang (2021) and Gao et al. (2021a) in this paper.

**FNO**: It is a powerful neural operator with FFT-based spectral convolutions (Li et al., 2020b). To make FNO have learned parameters that are of comparable scale as PICL, we employ FNO with 5 layers, and the width is 48.

**FNO\***: Based on FNO, we modify it to a similar architecture like PICL, which attaches an encoding module before the FNO model. So that, we can create a more level playing field for evaluating the effectiveness of our proposed method against the modified baseline, thereby providing a clearer measure of our contributions.

**PINO\***: PINO (Li et al., 2021) is a hybrid approach incorporating data and physics constraints based on FNO to learn the neural operator. We also introduced a specific modification, like what we do in FNO\*, to better address the nuances of our research problem, called PINO\*. The input and output are in coarse-grained mesh, which is used to calculate the physics loss. By using this baseline, we compare which is better to calculate physics loss on learnable fine-grained state or coarse-grained observation.

**LatentNeuralPDEs**: It is a space-time continuous grid-independent model for learning PDE dynamics from noisy and partial observations (Iakovlev et al., 2023). We employ the official implementation and compare with our proposed PICL.

## D  LIMITATIONS

The primary objective of this research is to establish a framework that integrates physics information into the training of models that rely only on coarse-grained data. This approach aims to improve the model's predictive capabilities in simulation or real-world problems. However, there are two notable areas in this work that present opportunities for further exploration.

Firstly, the framework is designed to be flexible in its use of the encoding and transition modules, not being limited to any fixed neural network architectures. In this study, we specifically explored the adaptation of a Transformer architecture as the encoding module. Future work could explore the efficacy of alternative neural network architectures for these modules.

Secondly, the current framework is tailored to situations where only coarse-grained data is available, without access to fine-grained data. We conducted some experiments and discussions on how fine-grained data (when available) improve the performance of PICL in this work. Investigating this aspect could provide deeper insights into the framework's applicability, particularly in scenarios where fine-grained data becomes accessible.

## E  COMPARE WITH LATENTNEURALPDES

Except for the FNO-based baselines, we also compare the PICL with other state-of-the-art methods like LatentNeuralPDEs (Iakovlev et al., 2023), and the results are shown in Table 9. We can see that the $\mathcal{L}_d$ of LatentNeuralPDEs is larger than those of PICL w/o fine-tune and PICL with fine-tune more than 81% and 91%, respectively.

Table 9: $\mathcal{L}_d$ of PICL and LatentNeuralPDEs based on NSWE.

|  | PICL w/o fine-tune | PICL with fine-tune | LatentNeuralPDEs |
|---|---|---|---|
| $\mathcal{L}_d$ | 3.69E-2 | **3.50E-2** | 6.70E-2 |

## F  EXPERIMENTS ON BURGERS EQUATION AND NAVIER-STOKES EQUATION

Except for the experiments on wave equation, linear shallow water equation, and nonlinear shallow water equation with uneven bottom, we also did the experiments on Burgers Eqn. and Navier-Stokes equation (NSE) following the work Li et al. (2020b). We compare the proposed PICL with baselines FNO\* and FNO, shown in Table 10.

Table 10: $\mathcal{L}_d$ of experiments on Burgers Eqn. and NSE.

| Method | Burgers Eqn. | NSE |
|---|---|---|
| FNO | 1.69E-2 | 1.06E-1 |
| FNO* | 1.51E-2 | 1.64E-2 |
| PICL w/o fine-tune | 1.39E-2 | 1.50E-2 |
| PICL with fine-tune | **1.38E-2** | **1.34E-2** |

From the Table 10, we can see that the $\mathcal{L}_d$ of PICL with fine-tune is slightly less than that of FNO* and FNO in the Burgers Equation experiment, while PICL with fine-tune has a significant improvement compared with FNO* and FNO in the experiment of NSE that is a more complex PDE than Burgers Equation.

## G  COMPUTATIONAL COST COMPARED WITH SOLVER

In this section, we explore the advantage of leveraging a neural network as the transition module instead of leveraging a numerical solver based on the computational cost. We calculate the computational time of a step forward by our transition module and a step computation of the numerical solver by FDM, same as that in Appendix B.1, to illustrate the computational cost. We do the experiments on different resolutions based on NSWE, including $32 \times 32, 48 \times 48, 64 \times 64$. The results are shown in Table 11. We can see that the time taken for a single computation using a numerical solver is more than 5 times that of a step forward with a neural network. Furthermore, as the resolution increases, the computational cost of using a neural network does not significantly change, as only the input and output layers are affected, with small changes in the hidden layers. These results indicate that, in terms of computational cost, whether in training or inference, employing a neural network in the transition module offers considerable advantages.

Table 11: Computational Time

| Method | $32 \times 32$ (s) | $48 \times 48$ (s) | $64 \times 64$ (s) |
|---|---|---|---|
| Neural Network | **0.0030** | **0.0032** | **0.0032** |
| Numerical Solver | 0.0159 | 0.0168 | 0.0175 |

## H  THE FULL RESULTS OF MULTI-STEP PREDICTIONS

The full results of the multi-step predictions for three benchmarks introduced in Sec. 5.2 & 5.3 & 5.4 is demonstrated in Tables 12.

## I  ADDITIONAL RESULT VISUALIZATION

In this section, we provide additional result visualizations for the Wave Eqn., LSWE, NSWE. The results of PINO*, FNO*, PICL with fine-tune, and ground truth corresponding to Table 1 are shown in Fig. 6. PICL with fine-tune (third column) can learn more accurate details than PINO* and FNO*. We use red circles to illustrate one of the improvements using PICL. For the Wave Eqn., there is an obvious improvement on $\phi$, like the different number of triangles in the circle, and the slight improvement on $u$ located near the edge. For the LSWE, there is the slight improvement on $h$, like the width in the middle of the saddle shape, an obvious improvement on $u^y$, like different shapes, and the slight improvement on $u^x$ like the size of the triangle at the edge. For the NSWE, there is an obvious improvement on $h$, like the middle dot, the slight improvement on $u^y$, like the width of the triangle at the edge, and an obvious improvement on $u^x$, like the link at the bottom.

Table 12: Performance of PICL and four baselines in the multi-step prediction of Wave Eqn., LSWE, and NSWE, measured by relative data loss $\mathcal{L}_d$. The first row is the steps predicted from the input in the range of 1 to 10 steps. We can see that our proposed PICL with fine-tune always has the best results shown as bold.

| Benchmarks | Steps | 1 | 2 | 3 | 4 | 5 | 6 | 7 | 8 | 9 | 10 |
|---|---|---|---|---|---|---|---|---|---|---|---|
| Wave Eqn. | PIDL | 1.36 | 2.41 | 3.27 | 4.10 | 4.76 | 5.36 | 6.07 | 6.68 | 6.96 | 7.09 |
| | PINO* | 1.18 | 2.00 | 2.54 | 2.91 | 3.01 | 2.94 | 2.84 | 2.80 | 3.01 | 3.46 |
| | FNO | 1.29E-1 | 1.36E-1 | 1.51E-1 | 1.76E-1 | 1.90E-1 | 2.00E-1 | 2.23E-1 | 2.73E-1 | 3.05E-1 | 3.34E-1 |
| | FNO* | 4.36E-2 | 7.99E-2 | 1.02E-1 | 1.15E-1 | 1.34E-1 | 1.58E-1 | 1.79E-1 | 2.04E-1 | 2.31E-1 | 2.53E-1 |
| | PICL w/o fine-tune | 2.74E-2 | 5.12E-2 | 6.91E-2 | 8.52E-2 | 1.05E-1 | 1.28E-1 | 1.54E-1 | 1.83E-1 | 2.10E-1 | 2.29E-1 |
| | PICL with fine-tune | **2.60E-2** | **4.87E-2** | **6.38E-2** | **7.57E-2** | **9.51E-2** | **1.19E-1** | **1.45E-1** | **1.70E-1** | **1.91E-1** | **2.07E-1** |
| LSWE | PIDL | 2.84 | 6.10 | 10.10 | 15.17 | 21.58 | 29.33 | 37.94 | 46.31 | 53.28 | 58.40 |
| | PINO* | 2.83 | 6.06 | 9.85 | 14.42 | 19.91 | 26.19 | 32.65 | 38.25 | 42.07 | 43.90 |
| | FNO | 4.13E-1 | 4.24E-1 | 4.32E-1 | 4.43E-1 | 4.58E-1 | 4.76E-1 | 4.99E-1 | 5.25E-1 | 5.55E-1 | 5.85E-1 |
| | FNO* | 2.18E-2 | 5.62E-2 | 9.71E-2 | 1.49E-1 | 2.16E-1 | 2.96E-1 | 3.84E-1 | 4.72E-1 | 5.54E-1 | 6.26E-1 |
| | PICL w/o fine-tune | 1.71E-2 | 3.13E-2 | 4.27E-2 | 5.92E-2 | 8.47E-2 | 1.18E-1 | 1.58E-1 | 1.99E-1 | 2.38E-1 | 2.71E-1 |
| | PICL with fine-tune | **1.41E-2** | **2.34E-2** | **3.53E-2** | **5.31E-2** | **7.80E-2** | **1.09E-1** | **1.45E-1** | **1.83E-1** | **2.20E-1** | **2.54E-1** |
| NSWE | PIDL | 2.27E-1 | 4.02E-1 | 5.40E-1 | 6.42E-1 | 7.10E-1 | 7.54E-1 | 7.86E-1 | 8.17E-1 | 8.53E-1 | 8.94E-1 |
| | PINO* | 2.24E-1 | 3.97E-1 | 5.32E-1 | 6.31E-1 | 6.97E-1 | 7.39E-1 | 7.68E-1 | 7.97E-1 | 8.32E-1 | 8.73E-1 |
| | FNO | 3.35E-1 | 3.44E-1 | 3.57E-1 | 3.74E-1 | 3.94E-1 | 4.10E-1 | 4.26E-1 | 4.53E-1 | 4.76E-1 | 4.87E-1 |
| | FNO* | 3.52E-2 | 7.34E-2 | 1.11E-1 | 1.48E-1 | 1.82E-1 | 2.12E-1 | 2.48E-1 | 2.92E-1 | 3.40E-1 | 3.91E-1 |
| | PICL w/o fine-tune | 1.75E-2 | 3.97E-2 | 6.67E-2 | 9.48E-2 | 1.18E-1 | 1.30E-1 | 1.37E-1 | 1.49E-1 | 1.70E-1 | 2.23E-1 |
| | PICL with fine-tune | **1.64E-2** | **3.29E-2** | **5.19E-2** | **7.36E-2** | **9.49E-2** | **1.10E-1** | **1.22E-1** | **1.40E-1** | **1.63E-1** | **1.82E-1** |

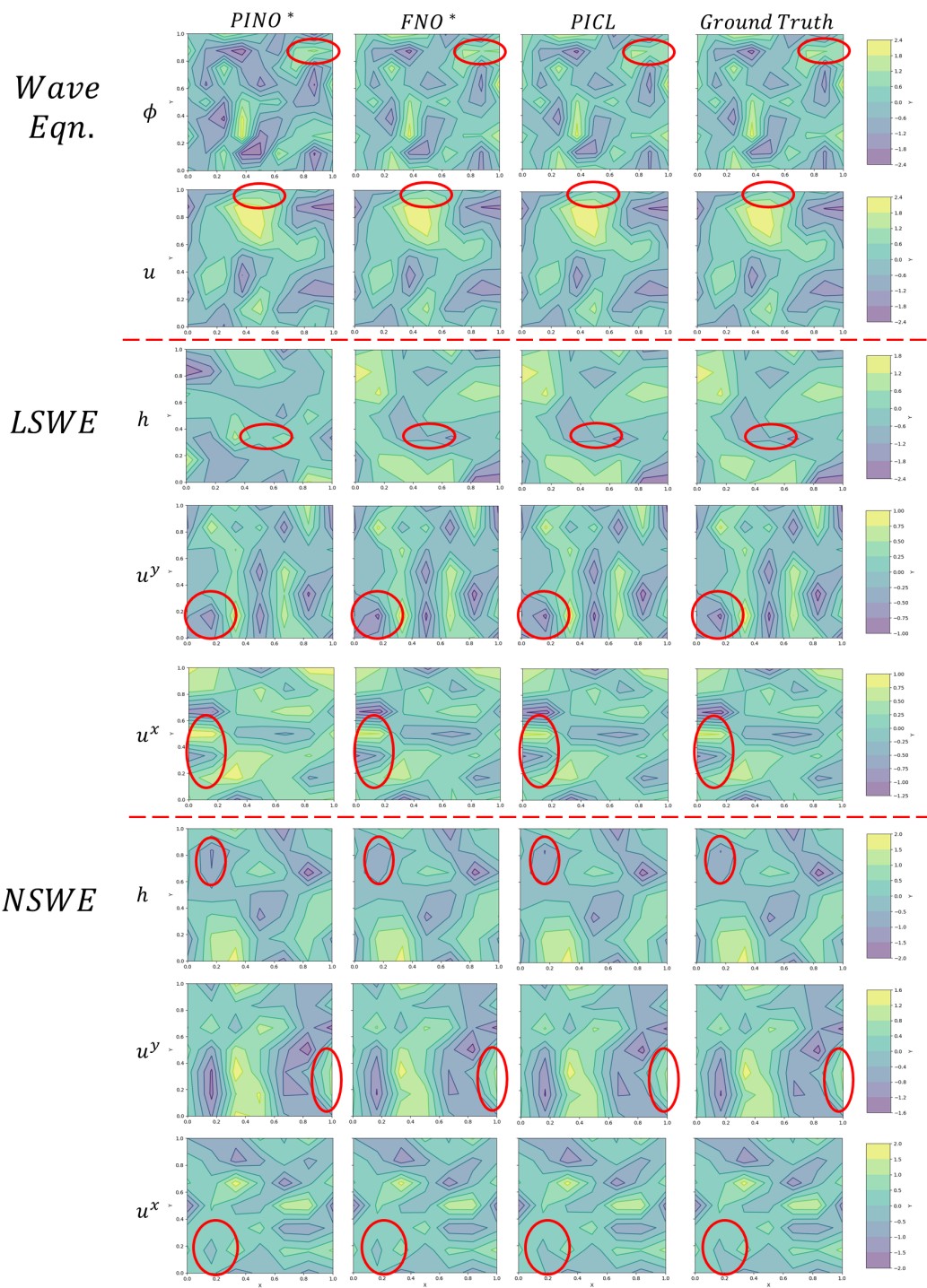

Figure 6: Visualization of predictions in (**Top**) Wave Eqn., (**Middle**) LSWE and (**Bottom**) NSWE experiments, using PINO*, FNO*, PICL, and ground truth. We use red circles to illustrate one of the improvements using PICL. **Wave Eqn.** $\phi$ denotes the field of velocity potential and $u$ denotes field of velocity. **LSWE** and **NSWE** $h$ denotes the field of fluid column height, $u^x$ and $u^y$ denote the fields of velocity in $x$ and $y$ directions. We can see that our PICL with fine-tune (third column) learns details better than PINO* and FNO*.

