# OpenReview forum: "PICL: Incorporating Coarse-Grained Data and Physics Information for Superior Physical Systems Modeling"
_ICLR.cc/2024/Conference — Submitted to ICLR 2024_

### Official Review · Reviewer_fJtB · 2023-10-28

**Soundness:** 2 fair
**Presentation:** 1 poor
**Contribution:** 1 poor
**Rating:** 3
**Confidence:** 4

**Summary:**

The authors introduced a framework to model coarse-grained data incorporating physics information in the form of partial differential equations (PDEs). The proposed framework learns to map coarse-grained data to fine-grained space and a dynamic operator that advances fine-grained states forward in time. The authors test the framework in a few exemplary PDEs and present some ablation studies on hyperparameters, data efficiency and quality.

**Strengths:**

* The authors explain the training routine thoroughly, which benefits reproducibility. Source code is also provided
* The writing is generally easy to follow, although held back by the lack of equations to explain key terms and concepts

**Weaknesses:**

* It is not very clear what the proposed framework is trying to achieve. The authors quotes "predictive ability" constantly but the prediction task is nowhere explicitly defined. I am inferring two tasks: (a) given coarse-grained state at time $t$, predict coarse-grained state at time $t+1$ and (b) given coarse-grained state, predict the corresponding fine-grained state. Task (a) inevitably involves incomplete information, which is not talked about and task (b) seems to be for a single step? (see questions below)
* The reason why the transition module is needed in the first place is unclear when there is already a closed expression for it from physics? It does not seem to provide benefits in terms of computational cost. If it represents approximate knowledge, the experiments do not demonstrate the value and limitations (compare against no physics knowledge at all and study how much error causes the framework to break down).
* The loss terms and metrics do not come with formulas and instead explained in dense text, making it difficult to follow

**Questions:**

* When the observations are extremely coarse-grained, they represent partial state and the map to fine-grained state is not one-to-one. How is this uncertainty accounted, especially given that the modules in the proposed framework are all deterministic in nature?
* Does evaluation involve only a single step prediction forward in time? Have you studied the performance over multiple steps?
* Broken sentence in section 4.1 below equation (3) - "the sequentially used transition module roles the prediction process..."

---

> ### Author Response · Authors · 2023-11-19
> **To Reviewer fJtB [Part 1]**
>
> Thank you for your valuable comments. We have conducted several additional experiments and comprehensively revised our paper to address your valuable questions and suggestions. Below are our detailed responses to each of your questions and concerns:
>
> > **Question about what does the prediction task mean and solution for incomplete information.**
>
> **A:** Thank you for your question.
>
> - Your inference 'a' is correct. In our paper, "predictive ability" refers to the capacity to predict the subsequent coarse-grained observations from the current coarse-grained observation. In light of your question, we have now included a clear definition and explanation of "predictive ability" in both the Abstract and the concluding paragraph of the Introduction.
> - For your mentioned incomplete information, we use coarse-grained observations from previous steps to estimate the current fine-grained state inspired by the finite difference method, to compensate incomplete information caused by coarse-grained observed data. We added the description discussion of our method in Section 4.2.1.
> -  Furthermore, for the impact of historical data length, we already did the ablation study and analyzed it in the third paragraph of Section 5.5.
>
> > **Question about why the transition module is required.**
>
> **A:** Thank you for your question. We also considered this aspect at the inception of our method's design, primarily for three reasons:
>
> - Firstly, numerous related studies, such as FNO [1], MAgNet [2], DINo [3], NeuralStagger [4], and others, focus on approximating numerical solvers using neural networks. Consequently, we decided to follow the ideas of these works, opting for neural network approximations rather than directly implementing numerical solvers.
> - Secondly, in response to your question, we have conducted an additional experiment to compare the computational costs of the neural network in the transition module versus a numerical solver, which demonstrates the value of the transition module. This comparative study is detailed in the Appendix G. The findings are as follows: The time taken for a single computation using a numerical solver is more than five times that of a step inference with a neural network. Furthermore, as the resolution increases, the computational cost of using a neural network does not significantly change, as only the input and output layers are affected, with small changes in the hidden layers. These results indicate that, in terms of computational cost, whether in training or inference, employing a neural network in the transition module offers considerable advantages.
> - Thirdly, employing a differentiable neural network is important for the future expansion of our proposed method. This approach may be beneficial for addressing control problems and inverse problems in subsequent research.
>
> [1] Li, Z., Kovachki, N., Azizzadenesheli, K., Liu, B., Bhattacharya, K., Stuart, A. and Anandkumar, A., 2020. Fourier neural operator for parametric partial differential equations. arXiv preprint arXiv:2010.08895.
>
> [2] Boussif, O., Bengio, Y., Benabbou, L. and Assouline, D., 2022. MAgnet: Mesh agnostic neural PDE solver. Advances in Neural Information Processing Systems, 35, pp.31972-31985.
>
> [3] Yin, Y., Kirchmeyer, M., Franceschi, J.Y., Rakotomamonjy, A. and Gallinari, P., 2022. Continuous pde dynamics forecasting with implicit neural representations. arXiv preprint arXiv:2209.14855.
>
> [4] Huang, X., Shi, W., Meng, Q., Wang, Y., Gao, X., Zhang, J. and Liu, T.Y., 2023. NeuralStagger: accelerating physics-constrained neural PDE solver with spatial-temporal decomposition. arXiv preprint arXiv:2302.10255.
>
> > **Suggestion in formulas of loss terms and metrics.**
>
> **A:** Thank you for your insightful suggestions. We have made revisions accordingly.
>
> - In Section 4.2.2, we have added more clearly definition each component of the loss function, including data loss and physics loss, using precise mathematical formulations.
> - Furthermore, in the first paragraph of Section 5, we have introduced a formal definition for the reconstruction error $\epsilon$ using a mathematical equation, replacing the previous textual description.

---

> ### Author Response · Authors · 2023-11-19
> **To Reviewer fJtB [Part 2]**
>
> > **Question about handling non-one-to-one mapping.**
>
> **A:** Thank you for your comments. I have clarified a few points regarding the non one-to-one mapping issue:
>
> - Due to the partially observed nature of the problem, we do not assume a one-to-one mapping between the current state $u_t$ and the next state $u_{t+1}$.
> - Instead, as described in Section 4.2.1, we use coarse-grained observations from previous steps to estimate the current fine-grained state. There are two interpretations of this: First, similar to PDEs, we can use the temporal derivative estimated from historical data to approximate the spatial derivative and reconstruct the current fine-grained state. Second, we assume a one-to-one mapping exists between a sequence of historical coarse-grained observations and the current fine-grained state.
> - We performed an ablation study (third paragraph, Section 5.5) analyzing the impact of the length of historical data used. This provides some empirical analysis of the mapping between observations and states.
> - As you pointed out, uncertainty-aware models like Bayesian neural networks and variational autoencoders are promising approaches for handling partial observations. I have added a discussion of these methods in Section 4.2.1.
>
> > **Question about multi-step performance.**
>
> **A:** Thank you for your question. Please note we have already show the multi-step prediction tests and compared them with baselines in our main paper. Please refer to Sections 5.2, 5.3, and 5.4, with results illustrated in Figure 2. To avoid confusion, we have revised the first paragraph of Section 5 by following your suggestion.
>
> > **Suggestion about sentence expression.**
>
> **A:** Thank you for your valuable suggestion. We have revised Section 4.2.1 to more accurately describe the transition module's role in the prediction phase. To be more clear, this sentence means the transition module can achieve the forecasting function. We design a neural network that inputs the current fine-grained state  $\hat{u}\_{t}$  to forecast the subsequent fine-grained state  $\hat{u}\_{t+1}$.

---

> ### Author Response · Authors · 2023-11-23
> **To Reviewer fJtB**
>
> Thank you again for your comments and suggestions to significantly improve the quality of our paper. We have read your comments carefully and understood your concerns. To address your concerns, we have revised several sections of our paper and conducted the additional experiments. Do you have any other concerns or suggestions?

---

### Official Review · Reviewer_iagL · 2023-10-30

**Soundness:** 3 good
**Presentation:** 2 fair
**Contribution:** 2 fair
**Rating:** 5
**Confidence:** 2

**Summary:**

The paper proposes a physics guided deep learning solution for modeling under data paucity and coarse-grained data. Essentially, the paper employs a super-resolution approach to generate fine-grained data from coarse-grained data employing supervised losses at the coarse-grained scale while employing physics losses (conservation conditions) between successive time-steps at the predicted fine-grained scale. Specifically, the proposed architecture comprises a sort of self-supervised task wherein the low-dimensional data is input into an encoder module which produces the corresponding high-dimensional output (predicted). This predicted high-dimensional output is passed into a transition module which predicts the high-dimensional output at the next time step. This high-dimensional output at the successive time-step is downsampled (by a deterministic function) and compared with the ground-truth low-dimensional data using a data-driven loss.

**Strengths:**

- The proposed solution is (somewhat) novel and is a creative way to effectively employ coarse-grained data and physics to perform super-resolution.


- The results are extensive (although not entirely convincing) and have been performed on multiple important PDE contexts.

**Weaknesses:**

1. Overall, the novelty in the paper is somewhat limited and analyses of the drawbacks of the proposed finite-difference based physics encoding method have not been fully carried out. Specifically, discussions regarding where the proposed method might lack, how fine-grained data can be incorporated (when available) will be helpful additions to the narrative.


2. Some results indicate that baselines outperform the proposed method. E.g., Table 1 NSWE indicate that PINO* has lower re-construction errors. Why is this?


3. The paper methodology is hard to understand and needs to be significantly improved. The reviewer feels the entire methodology can be explained in 1 – 2 paragraphs (half a page) but is needlessly convoluted and interspersed with details making it hard to get a high-level idea.


4.There are many ambiguous phrases/ design decisions that have been made without explanation.

    a. Why has the U-Net architecture been employed for the encoder while UNet++ [1] , Transformer [2] and many newer image encoding / SR architectures superior to UNet have been proposed more than 2 – 3 years ago?


    b. What does “hard encoding” $\tilde{o}$ into the corresponding $\hat{u}_t$ mean?

        i. Does it mean that assuming the low-res data was n/2 X h/2 and high-res data was n x h , that every 4th pixel in  the high-res data would have the corresponding $\tilde{o}$ value? Or does it mean something else?

        ii. If it means the same as <4.b.i>, would this design decision not overtly couple the high-res and low-res solutions? How might the high-res solution significantly improve upon the low-res solution with this constraint?


5. Results don't seem practically usable. It is important to comment on this owing to the context (i.e., mapping from low-res to high-res with predominantly low-res training data). In most real-world scientific simulations, physical consistency / errors are assumed to in the range `1e^-5 – 1e^-7` . A discussion about the practicality of the obtained results and the usability of the proposed method is required but missing.



References:

1. Zhou, Zongwei, et al. "Unet++: A nested u-net architecture for medical image segmentation." Deep Learning in Medical Image Analysis and Multimodal Learning for Clinical Decision Support: 4th International Workshop, DLMIA 2018, and 8th International Workshop, ML-CDS 2018, Held in Conjunction with MICCAI 2018, Granada, Spain, September 20, 2018, Proceedings 4. Springer International Publishing, 2018.


2. Dosovitskiy, Alexey, et al. "An image is worth 16x16 words: Transformers for image recognition at scale." arXiv preprint arXiv:2010.11929 (2020).

**Questions:**

1. Why has the U-Net architecture been employed for the encoder while UNet++ [1] , Transformer [2] and many newer image encoding / SR architectures superior to UNet have been proposed more than 2 – 3 years ago?


2. What does “hard encoding” $\tilde{o}$ into the corresponding $\hat{u}_t$ mean?

    a. Does it mean that assuming the low-res data was n/2 X h/2 and high-res data was n x h , that every 4th pixel in  the high-res data would have the corresponding $\tilde{o}$ value? Or does it mean something else?

    b. If it means the same as <2.a>, would this design decision not overtly couple the high-res and low-res solutions? How might the high-res solution significantly improve upon the low-res solution with this constraint?


3. Additionally, the function of $f_\theta$ in equation (1) is described as “imitating the implementation of higher-order finite difference to leverage abundant temporal feature of $\{ \tilde{o}_{t-i}\}_{i=0}^n$. ”

    a. What exactly does “imitating the implementation of higher-order FD” mean? Is there a FD operator that has been embedded into $f_\theta$? Or is there something special (I.e., some special input transformation) that has been applied to the inputs of $f_\theta$ that makes it “immitate” an FD operator?

---

> ### Author Response · Authors · 2023-11-19
> **To Reviewer iagL [Part 1]**
>
> Thanks for your reply and suggestions to improve our work. We have conducted several additional experiments and comprehensively revised our paper to address your valuable questions and suggestions. Below are our detailed responses to each of your questions and concerns:
>
> > **Concern about limitations.**
>
> **A:** Thanks for your question.
>
> - For the limitation of our proposed method: In Appendix D of our paper, we discuss the limitations of our work.
> - The first one is the compatibility of our proposed framework and the specific neural network architecture. We have conducted addtional experiments and made a detailed discussion in the 11st paragraph of Appendix B.6.
> - The second one is how to incropreate the avalable fine-grained data. We have conducted addtional experments and made a detailed discussion in the last paragraph of Appendix B.6. You may also check the following Question and Answer.
>
>
> > **Questions about the usage of fine-grained data.**
>
> **A:** Thanks for your question.
>
> - For aspect of how fine-grained data can be incorporated into our framework:
>   - Our current work is primarily focused on scenarios where only coarse-grained data is available, aligning with a significant portion of both simulation or real-world applications.
>   - We conjecture that if we assume the fine-grained data is available, the predictive performance will be better.
>   - To verify the above conjecture, we conducted an additional experiment incorporating fine-grained data. We detailed the experiment and its results in last paragraph of Appendix B.6. In this experiment, fine-grained data was used as the label for the encoding module, aiming to train the module to reconstruct a more accurate learnable fine-grained state. Our findings show that both the baseline FNO* (without physics loss) and our proposed PICL benefit significantly from the inclusion of fine-grained data, with improvements of over 38% and 12% respectively.
>   - Moreover, these results also highlight an important aspect of our framework: the ability to use the known physics information as a substitute for fine-grained data in training the encoding module, especially in scenarios where such data is unavailable. The smaller improvement observed in PICL compared to the FNO*, when fine-grained data is available, suggests that PICL can effectively compensate for the absence of fine-grained data to learn the state. Such discussion is also placed in Appendix B.6.
>
> > **Question about why some results of baselines outperform our method.**
>
> **A:** Thank you for pointing this issue.
>
> - For $\epsilon$ of PINO* outperforms PICL in Table 1, based on our experimental log data, we realized that there is a typo. The value should be 3.18E-1 instead of 3.18E-3. We have re-run the experiment multiple times, confirming that the magnitude is indeed in the E-1 range. We have fixed it in the table and extend our gratitude for your attention to this clerical error in our paper.
> - To avoid you from being confused. In Table 1, the PIDL's $\epsilon$ for LSWE performs better than PICL. We have explained it in Section 5.3, before. Even though the $\epsilon$ value in PIDL is slightly lower, this approach concentrates on training using only the physics loss, overlooking the importance of data constraints in predictions. Consequently, PICL has been developed to trade off between data and physics constraints, thereby improving the model's predictive ability. We consider this current observation to be normal, as our method aims to enhance the model's predictive ability using physical information in coarse-grained data, which is reflected in the $\mathcal{L}_{d}$ in our paper. The $\epsilon$ is only a metric to show the accuracy of the reconstructed fine-grained state, a step in our process, but not the end goal of our work.
>
> > **Suggestion about the lengthy introduction of methodology.**
>
> **A:** Thank you for your valuable suggestion, which has significantly contributed to the clarity and readability of our paper. We have revised Section 4 accordingly :
>
> - Firstly, we have succinctly introduced our concept and methodology in Section 4.1, using clear and easily understandable language to provide an overall view. This approach ensures that readers can quickly grasp the essence of our work.
> - Secondly, for those readers interested in the technical specifics, we delve deeper into the details of our method in the subsequent Section 4.2.

---

> > ### Author Response · Authors · 2023-11-19
> > **To Reviewer iagL [Part 2]**
> >
> > > **Question about why we employed U-Net as encoding module.**
> >
> > **A:** Thank you for your question.
> >
> > - As explained in the last paragraph of Section 4.2.1, we employed the U-Net as the encoding module, because the encoding is similar to this a super-resolution task and the U-Net is a proven model well-suited in super-resolution [1,2]. Importantly, our framework is not confined to any specific encoding or transition module.
> > - To address your question, we have conducted an new experiment to use a Transformer architecture as our encoding module, utilizing the official implementation of ViT [3].
> > - The details of this experiment and its results are presented in Appendix B.6, Paragraph 11, titled “Does the encoder can be replaced by other architecture?”. The findings demonstrate that our method enhances predictive capabilities over data-driven baselines, irrespective of whether the UNet or Transformer is employed as the encoding module. Moreover, the results indicate that using UNet yields better performance compared to the Transformer.
> >
> > [1] Ronneberger, O., Fischer, P. and Brox, T., 2015. U-net: Convolutional networks for biomedical image segmentation. In Medical Image Computing and Computer-Assisted Intervention–MICCAI 2015: 18th International Conference, Munich, Germany, October 5-9, 2015, Proceedings, Part III 18 (pp. 234-241). Springer International Publishing.
> >
> > [2] Esmaeilzadeh, S., Azizzadenesheli, K., Kashinath, K., Mustafa, M., Tchelepi, H.A., Marcus, P., Prabhat, M. and Anandkumar, A., 2020, November. Meshfreeflownet: A physics-constrained deep continuous space-time super-resolution framework. In SC20: International Conference for High Performance Computing, Networking, Storage and Analysis (pp. 1-15). IEEE.
> >
> > [3] Dosovitskiy, A., Beyer, L., Kolesnikov, A., Weissenborn, D., Zhai, X., Unterthiner, T., Dehghani, M., Minderer, M., Heigold, G., Gelly, S. and Uszkoreit, J., 2020. An image is worth 16x16 words: Transformers for image recognition at scale. arXiv preprint arXiv:2010.11929.
> >
> > > **Question about what does hard encoding mean and if the high resolution data is helpful.**
> >
> > **A:** Thank you for your question.
> >
> > - To address your concerns, we have clarified the concepts of down-sampling and hard encoding in Appendix B.5 of our paper, ensuring a more explicit explanation. Your understanding i is on point, and we will respond to question ii based on this understanding.
> > - We think this question as closely related to the second aspect of your initial comment regarding "Questions about the usage of fine-grained data." As mentioned previously, our current focus is on scenarios where physics information is used to enhance the model trained solely with coarse-grained data. In line with your suggestion, we extended an experiment involving high-resolution (fine-grained) data, and put it in last paragraph of Appendix B.6.
> > - This experiment confirmed that high-resolution data, when available, can indeed enhance the model's predictive capabilities. This is because if such high-resolution data is used as a label, the encoding module can reconstruct a more accurate state, which facilitates the prediction process in the transition period.

---

> > > ### Author Response · Authors · 2023-11-19
> > > **To Reviewer iagL [Part 3]**
> > >
> > > > **Concern about practice and real-world problems.**
> > >
> > > **A:** Thank you for your question. We have already considered this question before, and it is natural for the following reasons:
> > >
> > > - Firstly, some real-world problems have the known PDEs with error and noise, such as a paper published in Science in 2020 [1] and a paper published in PNAS 2023 [2]. They have demonstrated the ability to learn velocity and pressure fields from the partial observation data, strictly rely on the known Navier-Stokes equation using the physics-informed manner.
> > > - Secondly, there are many research questions in previous works [3-5] that are consistent with our problem, all assuming the PDEs are known, and conducting method research under this assumption.
> > >
> > > Your question is very interesting, we are also very intested in applying our framework on the real-world problem in the future.
> > >
> > > [1] Raissi, M., Yazdani, A. and Karniadakis, G.E., 2020. Hidden fluid mechanics: Learning velocity and pressure fields from flow visualizations. Science, 367(6481), pp.1026-1030.
> > >
> > > [2] Boster, K.A., Cai, S., Ladrón-de-Guevara, A., Sun, J., Zheng, X., Du, T., Thomas, J.H., Nedergaard, M., Karniadakis, G.E. and Kelley, D.H., 2023. Artificial intelligence velocimetry reveals in vivo flow rates, pressure gradients, and shear stresses in murine perivascular flows. Proceedings of the National Academy of Sciences, 120(14), p.e2217744120.
> > >
> > > [3] Li, Z., Zheng, H., Kovachki, N., Jin, D., Chen, H., Liu, B., Azizzadenesheli, K. and Anandkumar, A., 2021. Physics-informed neural operator for learning partial differential equations. arXiv preprint arXiv:2111.03794.
> > >
> > > [4] Wang, S., Wang, H. and Perdikaris, P., 2021. Learning the solution operator of parametric partial differential equations with physics-informed DeepONets. Science advances, 7(40), p.eabi8605.
> > >
> > > [5] Ren, P., Rao, C., Liu, Y., Ma, Z., Wang, Q., Wang, J.X. and Sun, H., 2023. PhySR: Physics-informed deep super-resolution for spatiotemporal data. Journal of Computational Physics, 492, p.112438.
> > >
> > > > **Confusion about imitating the FD.**
> > >
> > > **A:** Thank you for pointing out the need for clearer language in our manuscript. We have revised the wording in the first paragraph of Section 4.2.1 to more precisely describe the meaning.
> > >
> > > - Specifically, we have changed the term "imitating" to "inspired by." Here, we aim to provide intuition for why we incorporate the historical coarse-grained observation as input data. More concretely, when we leverage the finite difference (FD) operators, we actually use the temporal finite difference to calculate the spatial finite difference. This means the historical data used to calculate the temporal finite difference can help estimate the (spatial) fine-grained state at the current time.
> > > - Moreover, we performed an ablation study analyzing the impact of historical data length, which is discussed in the third paragraph of Section 5.5.

---

> > > > ### Comment · Reviewer_iagL · 2023-11-22
> > > > **Response to Authors**
> > > >
> > > > Thank you to the authors for their detailed response. Although some of the questions have been answered and despite some improvement in the clarity of the proposed method, the reviewer still feels (i) the main paper cannot stand on its own without frequent references to the appendix even for description of the proposed model (e.g., details about two-stage fine-tuning, details about discretization function etc.) (ii) the motivation for a solution employing coarse-grained modeling (with intermediate "fine-grained" predictions only controlled by physics losses) is unclear to the reviewer i.e., how could the quality of the "super-resolved" intermediate predictions in the proposed model be any better than enabled by the low-res data (i.e., the upper-limit of resolution is controlled by low-res data so the "fine-grained" estimates are just larger "coarse-grained" estimates) especially when there is no fine-grained data (even a few instances) included during model training? This question is still unclear to the reviewer (bringing into question the motivation of the formulation) and a lack of detailed discussion about limitations from this perspective is still a direction that requires major improvement in the paper.
> > > >
> > > > The reviewer understands that the authors did execute an updated experiment with fine-grained data but the motivation for the original problem formulation (i.e. ,the entire reason for the current model design is assuming only availability of coarse-grained data) is unclear and the reasoning behind the improved performance just using intermediate physics losses of this design is unclear. Specifically, what is preventing the model from learning a simple interpolation in the "fine-grained" space of the coarse-grained inputs. Often the fine-grained simulations (if the resolution is 4x or more than the coarse-grained simulations) have different local signatures not present in the coarse-grained case. How can the model learn to represent these signatures (in the fine-grained space), when it has never seen them, solely by learning from the  coarse-grained inputs and the physics losses?
> > > >
> > > > Owing to (i), (ii) the reviewer shall retain the score at 5 for this paper.

---

> > > > > ### Author Response · Authors · 2023-11-22
> > > > > **To Reviewer iagL**
> > > > >
> > > > > Thanks for your reply. We are happy to hear our efforts have addressed some of your concerns. We believe your remaining concerns focus on two main points:
> > > > >
> > > > > - The main paper lacks self-contained detail and currently relies heavily on the appendix in Section 4.
> > > > > - The motivation for the coarse-grained modeling and effectiveness of the fine-grained prediction module.
> > > > >
> > > > > > **The main paper lacks self-contained detail and currently relies heavily on the appendix in Section 4.**
> > > > >
> > > > > - We have revised Sec. 4.2.2 to describe the details of the two-stage fine-tuning period.
> > > > > - We have revised Sec. 4.2.1 to introduce the definition and details of the hard-encoding operation in the main paper.
> > > > > - We have also revised other parts, including the details of the physics loss, so that the description of the proposed model can be easier to understand without the appendix.
> > > > >
> > > > > > **The motivation of the coarse-grained modeling and the effectiveness of the fine-grained prediction module.**
> > > > >
> > > > > - The objective of our work is not to build low-resolution surrogate models, but to develop a method for better prediction when limited to coarse-grained observations. In some cases, prediction and control rely exclusively on such observations. Our main idea is to incorporate physics information into the modeling system to improve its performance.
> > > > > - Our key motivation is the consensus that incorporating physical priors improves model performance, as evidenced in prior works like PINO [1], Physics-informed DeepONets [2], and PhySR [3]. Why this occurs remains an open theoretical question. One explanation is that physical priors reduce the hypothesis space, enabling better generalization.
> > > > > - However, given only coarse-grained data, leveraging physics to constrain models remains challenging. The key contribution of our work is providing a method to overcome this.
> > > > > - In an intuitive sense, the formulation of the Partial Differential Equation (PDE), such as $$\frac{\partial u}{\partial t} = \frac{\partial u}{\partial x},$$ delineates a relationship between temporal and spatial derivatives. This implies that there exists a potential to utilize temporal data from preceding steps to reconstruct spatial information at the current time step, by harnessing the principles of physics equations. It's important to note that the state we are dealing with is not a typical natural image, but a state governed by a PDE.
> > > > > - Specifically, we conducted an additional experiment demonstrating our method learns superior fine-grained predictions compared to methods without physics loss or interpolation. $\epsilon$ is the relative MSE of the fine-grained prediction and real fine-grained data.
> > > > >
> > > > > | Methods | $\epsilon$ |
> > > > > |---------|---------|
> > > > > | Model without physics loss   | 1.74   |
> > > > > | Nearest neighbors interpolation   | 8.40E-1   |
> > > > > | Bilinear interpolation  | 5.98E-1   |
> > > > > | Bicubic interpolation   | 5.83E-1   |
> > > > > | Model with PICL   | **2.03E-1**  |
> > > > >
> > > > > [1] Li, Z., Zheng, H., Kovachki, N., Jin, D., Chen, H., Liu, B., Azizzadenesheli, K. and Anandkumar, A., 2021. Physics-informed neural operator for learning partial differential equations. arXiv preprint arXiv:2111.03794.
> > > > >
> > > > > [2] Wang, S., Wang, H., & Perdikaris, P. (2021). Learning the solution operator of parametric partial differential equations with physics-informed DeepONets. Science advances, 7(40).
> > > > >
> > > > > [3] Ren, P., Rao, C., Liu, Y., Ma, Z., Wang, Q., Wang, J.X. and Sun, H., 2023. PhySR: Physics-informed deep super-resolution for spatiotemporal data. Journal of Computational Physics, 492, p.112438.

---

### Official Review · Reviewer_vB1V · 2023-10-30

**Soundness:** 3 good
**Presentation:** 3 good
**Contribution:** 2 fair
**Rating:** 3
**Confidence:** 5

**Summary:**

In this paper, the author proposes a new physics-informed framework for achieving finer-grained data reconstruction in spatial and temporal fields from coarser-grained data through super-resolution and neural operators. The framework comprises two components: an encoder module and a transition module, and it utilizes a three-stage training process (base training followed by a two-stage fine-tuning process). The results show the effectiveness of the proposed method in different PDE-governed systems. However, it lacks some important comparisons with non-FNO-based approaches and also some discussion about details of incorporating physics.

**Strengths:**

1.	The paper studies an important problem. The authors well introduce the background and define the problem while providing a clear overview of the proposed framework.
2.	The authors justifies the effectiveness of the proposed method in various aspects through a series of experiments over multiple PDE-governed systems.

**Weaknesses:**

1.	The method section lacks a clear explanation of how the physics loss is defined and how the 4th-order Runge-Kutta (RK) method is incorporated into the design.
2.	The experimental section only compares the method with Fourier Neural Operator (FNO)-based methods, omitting comparisons with other state-of-the-art neural operator methods, such as similar mesh-based methods like Magnet.
3.	The original FNO paper tested model performance on different Partial Differential Equations (PDEs), including 1D cases (decay flow) and 2D cases (Navier-Stokes equations). It would be beneficial if the author also compared their method against these PDEs.
4.	The performance in zero-shot super-resolution (SR) cases is not addressed.

**Questions:**

See weaknesses points above.

---

> ### Author Response · Authors · 2023-11-19
> **To Reviewer vB1V [Part 1]**
>
> Thanks for your recognition of our definition of the problem. We have conducted several additional experiments and comprehensively revised our paper to address your valuable questions and suggestions. Below are our detailed responses to each of your questions and concerns:
>
> > **Question about how physics loss is defined.**
>
> **A:** Thank you for your valuable feedback regarding our experiments.
>
> - According to your question, we have added a more detailed explanation immediately following Eqn. 5 in Section 4.2.2. The enhancement includes an in-depth description of the physics loss and the introduction of the RK4 method are placed in Appendix B.2.
> - We first present the process of RK4, which can be considered as a function $F(u_t,u_{t+1})=0$. By using $F$, we can design two physics losses $\mathcal{L}\_{ep}(\theta)=F(u_t(\theta), u_{t+1}(\theta))^2$  for the encoding module and $ \mathcal{L}\_{tp}(\omega) = F(u_t, u_{t+1}(\omega))^2$ for the transition module. By minimizing the $\mathcal{L}\_{ep}$ and $\mathcal{L}\_{tp}$, the $u_t$ and $u\_{t+1}$ can be found, which means that the $u\_t$ and $u\_{t+1}$ can satisfy the RK4 time differentiation.
>
> > **Suggestion of adding baseline.**
>
> **A:** Thank you for your suggestion.
>
> - Because FNO and PINO are common methods that many works about neural operator used as baselines, we previously compared our method with them.
> - Thank you for mentioning MAgNet. We have done further literature review and found the work of three state-of-the-art neural operators, including MAgNet [1], DINo [2], and LatentNeuralPDEs [3], and cited them in the Related Work. Considering the limitations of computing resources, we selected the method LatentNeuralPDEs with the best performance as our baseline.
> - We have compared our proposed method with the official implementation of LatentNeuralPDEs. The details about LatentNeuralPDEs are described in Appendix C, and the comparative results are presented in Appendix E and Table 9. The findings indicate that the loss ($\mathcal{L}_{d}$) of LatentNeuralPDEs is larger than that of PICL w/o fine-tune and PICL with fine-tune by more than 81% and 91%, respectively.
>
> [1] Boussif, O., Bengio, Y., Benabbou, L. and Assouline, D., 2022. MAgnet: Mesh agnostic neural PDE solver. Advances in Neural Information Processing Systems, 35, pp.31972-31985.
>
> [2] Yin, Y., Kirchmeyer, M., Franceschi, J.Y., Rakotomamonjy, A. and Gallinari, P., 2022. Continuous pde dynamics forecasting with implicit neural representations. arXiv preprint arXiv:2209.14855.
>
> [3] Iakovlev, V., Heinonen, M. and Lähdesmäki, H., 2023, November. Learning Space-Time Continuous Latent Neural PDEs from Partially Observed States. In Thirty-seventh Conference on Neural Information Processing Systems.
>
> > **Suggestion of more PDEs experiments.**
>
> **A:** Thank for your suggestion. We agree that additional experiments involving PDEs used in FNO are essential. Regarding your question about the 1D case decay flow, you might mean the Burgers flow experiment in FNO, as decay is a characteristic feature of the viscous Burgers equation? If not, please tell me what you mean.
>
> - Based on your suggestion, we have conducted experiments on the Burgers Equation and the Navier-Stokes Equation following the settings in original FNO paper, and compared the results with FNO. The detailed experimental setup and results are presented in the Appendix F of our paper.
> - We summarize the results as follows: In the Burgers Equation experiments, the data loss ($\mathcal{L}_{d}$) of PICL with fine-tuning is marginally lower than that of FNO* and FNO. In the Navier-Stokes Equation experiment, which involves a more complex PDE, PICL with fine-tuning demonstrates significant improvement over both FNO* and FNO. These results are detailed in Table 10. This enhancement in performance shows the efficacy of our approach and its potential in handling intricate fluid dynamics problems.

---

> > ### Author Response · Authors · 2023-11-19
> > **To Reviewer vB1V [Part 2]**
> >
> > > **Concern about addressing zero-shot super-resolution cases.**
> >
> > **A:** Thank you for your insightful question. Initially, our work did not focus on addressing the issue of zero-shot super-resolution. Inspired by your questions, we have conducted additional experiments to assess the zero-shot super-resolution capabilities of our proposed framework.
> >
> >
> > - The detail of experiment is descripted in the 13rd paragraph of Appendix B.6 named “How to address the zero-shot super-resolution cases?”. The conclusion is: The data presented in the Fig. 5 clearly demonstrates that the PICL with fine-tune consistently outperforms the baseline FNO*, regardless of the inference sizes of coarse-grained meshes. This indicates that the incorporation of physics information can significantly enhance the zero-shot super-resolution of the encoding module. This improvement is achieved by ensuring that the output adheres to the underlying physics equation, a crucial factor especially when fine-grained data is not available.
> > - Here is how we modified the network: The original UNet encoding module requires a consistent coarse-grained meshes between inference and training, making it difficult to complete the zero-shot super-resolution task. The goal of our work is to develop a training framework that is not limited to specific encoding module. Grateful for the inspiration provided by MAgNet, we apply the nearest neighbors interpolation as same as MAgNet in the encoding module. Then, by replacing the UNet to Transformer as encoding module, our proposed framework can address the zero-shot super-resolution cases.

---

> ### Author Response · Authors · 2023-11-23
> **To Reviewer vB1V**
>
> Thank you again for your comments and suggestions to significantly improve the quality of our paper. We have read your comments carefully and understood your concerns. To address your concerns, we have revised several sections of our paper, and conducted numerous experiments on the additional baseline, different PDEs, and zero-shot super-resolution cases. Do you have any other concerns or suggestions.

---

### Official Review · Reviewer_vvjE · 2023-10-30

**Soundness:** 3 good
**Presentation:** 2 fair
**Contribution:** 2 fair
**Rating:** 5
**Confidence:** 3

**Summary:**

The paper addresses an issue typically seen in real world problems, where only coarse-grained measurements are available. The authors propose a surrogate model that works over data with finer resolution reconstructed from observed data with low resolutions. The model comprises U-Net and Fourier neural network models and the proposed training framework incorporates physical losses that ensure consistency between reconstructed solutions.

**Strengths:**

The paper is motivated by real world applications, to which few surrogate physics models have been applied so far. The authors propose a novel framework that incorporates losses stemmed from physical systems. The authors also propose two training phases, where the first phase is responsible for training models in a supervise manner and the second phase is based on fine-tuning aimed for enhancing the prediction capability. The experiments show that the proposed method outperforms a couple of baselines. Some ablation studies are also conducted.

**Weaknesses:**

I do not quite understand the problem setting in the paper yet. What problems does the paper aim to tackle? To my knowledge, the prediction of coarse-grained observation is not a keen requirement for real world applications since simulation in real applications is mostly performed with high resolution data.


What is the difference between PICL and FNO* reported in the experiments? Are the loss functions used in training different? It is also unclear how much each term of the loss function has an impact to the performance of the proposed forward model.


When the output resolution of $f_{\theta}$ increases, how much does it have an impact to the computational cost? How does it compare to the baselines?


The paper does not seem to be well-organized and hard to follow. The followings are some of the examples that made the paper difficult to understand:
* Definition in Section 3 is ambiguous. For instance, differential operator $P$ is not well-defined since Banach spaces $(A, U, V)$ is not mentioned to have differential structure.
* Between equations (3) and (4),  "Then, the sequentially used transition module roles the prediction process with the input $u^{t}$ to predict $u^{t+1}$".
* I could not find the implementation detail of down-sampling operator $\Phi$.
* Figure 1 is very confusing since arrows corresponding to data flow in training and test phases are mixed up. Are physics loss $L_{ep}$ and $ L_{tp}$ used in the unrolling phase?

**Questions:**

The proposed method relies on physics loss defined by fourth-order central-difference scheme and Runge-Kutta time discretization, which can be performed when one knows PDEs of the problems. Do the authors have any observation or insight on what range of real world problems the proposed method work for?

**Details Of Ethics Concerns:**

I don't have any concerns.

---

> ### Author Response · Authors · 2023-11-19
> **To Reviewer vvjE [Part 1]**
>
> Thank you for your valuable reply. We have conducted several additional experiments and comprehensively revised our paper to address your valuable questions and suggestions. Below are our detailed responses to each of your questions and concerns:
>
> > **Question about the problem setting and its importance.**
>
> **A:** Thank you for your question. To make our paper clearer and easier to follow, we have made the following adjustment to the paper:
>
> - For the problem setting, the scientific problem we solved is how to apply physics information on coarse-grained data to improve predictive ability, whether in numerical simulations or real-world applications when we only have coarse-grained data. We revised the Abstract and the 2nd paragraph of the Introduction to make it clear.
> - For its importance:
>   - In real-world scenarios, only coarse-grained data measured by sensors is available that is the importance of the problem setting.
>   - Generally, this framework can be applied to any problem seeking to enhance model predictions using physics information on coarse-grained data, offering insights and potential solutions.
>
> > **Differences between PICL, FNO\*, and impacts of each term in loss.**
>
> **A:** Thank you for your question.
>
> - We have made revisions to Section 5.1, first paragraph, to provide a more detailed description to each baseline for better understanding. Here to clarify: To control variables, PICL and the FNO* baseline are very similar in terms of network structure. Specifically, the primary difference between PICL and FNO* is that FNO* omits the physics loss components from the PICL’s loss function, retaining only the data loss.
> - Regarding the impact of each term in loss functions, our paper already discussed this in Section 5.5 Ablation Studies, specifically in the second paragraph. Our experimental findings indicate that including a physics loss in the loss function leads to better results compared to method such as FNO* without it. The model performs optimally when the physics loss coefficient, $\gamma$ = 1E-1. This is because a very low weight for physics loss makes it ineffective in the optimization process, while a very high weight causes the model to overly prioritize satisfying physical constraints at the expense of predictive accuracy. We appreciate you highlighting this issue, and we have made revisions to this section for greater clarity in describing the ablation study.
>
> > **Concern about computational cost.**
>
> **A:** Thank you for raising this intriguing question.
>
> - In response, we have added a group of experiment to Appendix B.6, specifically in the 9th paragraph, titled "When the output resolution of the encoder module increases, how much does it impact the computational cost?"
> - Our findings are as follows: During inference, as the resolution of the encoder module's output increases from $32\times32$ to $48\times48$, and then to $64\times64$, the computational time (s) for the baseline without physics loss increased by approximately 92% and 65%, respectively. In comparison, PICL saw an increase of about 83% and 69%. The growth in computational cost for both methods is quite similar. During training, the baseline without physics loss increased by about 277% and 79%, while PICL increased by approximately 238% and 48%. Although PICL's training time increased, its computational cost grew less than the baseline with the rise in resolution. In summary, when using PICL for inference, the computational cost considers only the time for inference. Hence, the efficiency of PICL is essentially comparable to the baseline without physics loss, as PICL does not require computation of physics loss during inference.

---

> > ### Author Response · Authors · 2023-11-19
> > **To Reviewer vvjE [Part 2]**
> >
> > > **Concern about paper’s organized.**
> >
> > **A:** Thank you for pointing out the organizational aspects of our paper. We have undertaken comprehensive revisions across various sections, including the abstract, introduction, preliminaries, and methodology.
> >
> > Specifically:
> >
> > - **Problem 1:** We appreciate your feedback and have reorganized the definitions of problems and operators in the preliminaries. The domain of the problem we want to solve is similar to existed works like FNO [1] and PINO [2], and it is also a prediction problem in the PDE-governed system. Therefore, we referred to their definition of the problem and revised the section Preliminaries to maintain consistency with these works. This includes a refined definition of the differential operator $P$, an enhanced discussion of their discrete representations, and the clear definition of down-sampling operation.
> > - **Problem 2:** We have revised Section 4.2.1 to more accurately describe the transition module's role in the prediction phase. To be more clear, this sentence means the transition module can achieve the forecasting function. We design a neural network that inputs the current fine\-grained state $\hat{u}\_{t}$ to forecast the subsequent fine-grained state $\hat{u}_{t+1}$.
> > - **Problem 3:** We have revised the last paragraph of the preliminaries for clarity and provided a detailed mathematical formulation and execution details of this operation in Appendix B.5. To be more clear, the down-sampling means we down-sample the state values in the fine-grained data $\tilde{u}$ and then rearrange them to a matrix that is the coarse-grained observation $\tilde{o}$.
> > - **Problem 4:** Thanks for your question. Before, we only show the training period in Figure 1. Now, we have added three figures to show the inference period, physics-tuning stage, and data-tuning stage in the paper, for the reader can easily understand. The figures of training period and inference period are placed in section 4.1, and the figures of physics-tuning stage and data-tuning stage in two-stage fine-tuning period are place in Appendix B.3.
> >
> > > **Question about real-world problems the proposed method work for.**
> >
> > **A:** Thank you for your question. We have already considered this question before, and it is natural for the following reasons:
> >
> > - Firstly, some real-world problems have the known PDEs, such as a paper published in Science in 2020 [1] and a paper published in PNAS 2023 [2]. They have demonstrated the ability to learn velocity and pressure fields from the partial observation data, strictly rely on the known Navier-Stokes equation using the physics-informed manner.
> > - Secondly, there are many research questions in previous works [3-5] that are consistent with our problem, all assuming the PDEs are known, and conducting method research under this assumption.
> >
> > Your question is very interesting, we are also very intested in applying our framework on the real-world problem in the future.
> >
> > [1] Raissi, M., Yazdani, A. and Karniadakis, G.E., 2020. Hidden fluid mechanics: Learning velocity and pressure fields from flow visualizations. Science, 367(6481), pp.1026-1030.
> >
> > [2] Boster, K.A., Cai, S., Ladrón-de-Guevara, A., Sun, J., Zheng, X., Du, T., Thomas, J.H., Nedergaard, M., Karniadakis, G.E. and Kelley, D.H., 2023. Artificial intelligence velocimetry reveals in vivo flow rates, pressure gradients, and shear stresses in murine perivascular flows. Proceedings of the National Academy of Sciences, 120(14), p.e2217744120.
> >
> > [3] Li, Z., Zheng, H., Kovachki, N., Jin, D., Chen, H., Liu, B., Azizzadenesheli, K. and Anandkumar, A., 2021. Physics-informed neural operator for learning partial differential equations. arXiv preprint arXiv:2111.03794.
> >
> > [4] Wang, S., Wang, H. and Perdikaris, P., 2021. Learning the solution operator of parametric partial differential equations with physics-informed DeepONets. Science advances, 7(40), p.eabi8605.
> >
> > [5] Ren, P., Rao, C., Liu, Y., Ma, Z., Wang, Q., Wang, J.X. and Sun, H., 2023. PhySR: Physics-informed deep super-resolution for spatiotemporal data. Journal of Computational Physics, 492, p.112438.

---

> ### Comment · Reviewer_vvjE · 2023-11-21
> **Thank you very much for the reply.**
>
> Thank you for your answers. Given the contributions of the paper as well as the willingness of the authors' to provide explanations and a considerable amount of additional experimental results, I raised my score to 5. While most of my concerns were addressed, it is still unclear the necessity that we learn surrogate models able to perform low-resolution simulation. The authors' assumption that only coarse-grained data is available in real-world applications is reasonable, but I still doubt that this assumption also motivates one to have low-resolution surrogate models, because in real applications high-fidelity simulation are much in demand.

---

> > ### Author Response · Authors · 2023-11-22
> > **To Reviewer vvjE**
> >
> > Thanks for your reply and happy to hear that our efforts can successfully address your concern. Besides, we would like to argue that our motivation is not to build a low-resolution surrogate that can perform low-resolution simulation.
> >
> > **A:** Our proposed framework and the coarse-grained surrogate boast a broad spectrum of applications pertinent to physical systems, such as fluid field control [1-3], ocean energy generalization [4,5], air pollution assessment [6], and weather prediction [7]. In these contexts, we can only acquire coarse-grained observations from sensors. In fact, in the majority of real applications, the data is coarse-grained as the dense deployment of sensors is impractical. Our objective is not to build low-resolution surrogate models for a system but to develop a method that leverages the physics information in circumstances where we only have access to coarse-grained observations. As evidenced by the aforementioned works, predictions and control can only be conducted based on coarse-grained observations. The problem addressed by our proposed framework was abstracted from these challenges and framed as a scientific issue.
> >
> >
> > [1] Paris, R., Beneddine, S. and Dandois, J., 2021. Robust flow control and optimal sensor placement using deep reinforcement learning. Journal of Fluid Mechanics, 913, p.A25.
> >
> > [2] Li, S., Li, W. and Noack, B.R., 2022. Machine-learned control-oriented flow estimation for multi-actuator multi-sensor systems exemplified for the fluidic pinball. Journal of Fluid Mechanics, 952, p.A36.
> >
> > [3] Castellanos, R., Cornejo Maceda, G.Y., De La Fuente, I., Noack, B.R., Ianiro, A. and Discetti, S., 2022. Machine-learning flow control with few sensor feedback and measurement noise. Physics of Fluids, 34(4).
> >
> > [4] Desouky, M.A. and Abdelkhalik, O., 2019. Wave prediction using wave rider position measurements and NARX network in wave energy conversion. Applied Ocean Research, 82, pp.10-21.
> >
> > [5] Jamei, M., Ali, M., Karbasi, M., Xiang, Y., Ahmadianfar, I. and Yaseen, Z.M., 2022. Designing a multi-stage expert system for daily ocean wave energy forecasting: A multivariate data decomposition-based approach. Applied Energy, 326, p.119925.
> >
> > [6] Cordova, C.H., Portocarrero, M.N.L., Salas, R., Torres, R., Rodrigues, P.C. and López-Gonzales, J.L., 2021. Air quality assessment and pollution forecasting using artificial neural networks in Metropolitan Lima-Peru. Scientific Reports, 11(1), p.24232.
> >
> > [7] Su, F., Fan, R., Yan, F., Meadows, M., Lyne, V., Hu, P., Song, X., Zhang, T., Liu, Z., Zhou, C. and Pei, T., 2023. Widespread global disparities between modelled and observed mid-depth ocean currents. Nature Communications, 14(1), p.2089.

---

### Author Response · Authors · 2023-11-20
**General Comments and Revision Summary**

We thank all reviewers for your careful review and constructive comments and suggestions. We also thank all reviewers for your recognition of the idea, method, and presentation of our work. We summarize our main revision of the paper below.

- According to the reviewer's suggestion, we added several experiments, including a new baseline LatentNeuralPDEs, the impact of output resolution on computational cost, the performance using the Transformer encoding module, the zero-shot super-resolution cases, the improvement by fine-grained data (when available), the Burgers Equation and Navier-Stokes Equation, and the comparison with solver on computational cost.
- We revised the Sec. 3 Preliminaries, Sec. 4 about our method, and Sec. 5 Experiments to make it easier to follow. Specifically, (1) we clarified the definition of the key concepts and variables in Sec. 3. (2) we first described the overview of our framework and problem setting using easily understandable language in Sec. 4.1 and introduced the details of our framework in Sec. 4.2. (3) we clarified the definition of physics loss with formulas in Sec. 4.2. (4) we clarified several descriptions of baselines in Sec. 5.
- We carefully read the related works provided by all reviewers and cite them properly in the paper. We compared with a state-of-the-art additional baseline.
- We revised the whole paper to enhance its readability and fixed the typos.
- Last but not least, we thank all reviewers again for the effort they put into this paper which makes this paper better.

---

### Meta-Review · Area_Chair_YLop · 2023-12-07

**Metareview:**

The paper explores the development of surrogate models for dynamical systems when only coarse-grained data is available. To achieve this, it incorporates physical priors and utilizes existing data information to model the underlying process dynamics at a fine-grained resolution. The proposed model follows an encode-process-decode approach. Experiments were conducted on three types of partial differential equations (PDEs).

The reviewers agree that the paper is not yet ready for publication. The objectives need clarification and better articulation, the paper's organization requires improvement, and the experimental section needs to be reinforced. Throughout the discussion, the authors enhanced the initial version of the paper through reorganization and additional experiments. While this effort was acknowledged by the reviewers, it did not suffice to change their overall assessment.

**Justification For Why Not Higher Score:**

Reviewers found that the paper is not ready for publication.

**Justification For Why Not Lower Score:**

a

---

### Decision · Program_Chairs · 2024-01-16

Reject